



# Exploring the Long-Term Reanalysis of Precipitation and the Contribution of Bias Correction to the Reduction of Uncertainty over South Korea: A Composite Gamma-Pareto Distribution Approach to the Bias Correction

Dong-Ik Kim[1], Hyun-Han Kwon[2] and Dawei Han[1]

[1]Water and Environment Research Group, Department of Civil Engineering, University of Bristol, United Kingdom
[2]Department of Civil Engineering, Chonbuk National University, South Korea

*Correspondence to*: Hyun-Han Kwon (hkwon@jbnu.ac.kr)

**Abstract.** The long-term record of precipitation data plays an important role in climate impact studies.
The local observation is often considered to be "the truth" in regional-scale analyses, but the long-term meteorological record for a given catchment is very limited. Recently, ERA-20c, a century-long reanalysis of the data has been published by the European Centre for Medium-Range Weather Forecasts (ECMWF), which includes daily precipitation over the whole 20th century with high spatial resolution of 0.125°× 0.125°. Preliminary studies have already indicated that the ERA-20c can reproduce the mean reasonably
well, but rainfall intensity was underestimated and wet-day frequency was overestimated. The primary focus of this study was to expand our sample size significantly for extreme rainfall analysis. Thus, we first adopted a relatively simple approach to adjust the frequency of wet-days by imposing an optimal lower threshold. We found that the systematic errors are fairly well captured by the conventional quantile mapping method with a gamma distribution, but the extremes in daily precipitation are still somewhat
underestimated. In such a context, we introduced a quantile mapping approach based on a composite distribution of a generalized Pareto distribution for the upper tail (e.g. 95th and 99th percentile), and a gamma distribution for the interior part of the distribution. The proposed composite distributions provide a significant reduction of the biases compared with that of the conventional method for the extremes. We suggest a new interpolation method based on the parameter contour map for bias correction in ungauged
catchments. The strength of this approach is that one can easily produce the bias-corrected daily precipitation in ungauged or poorly gauged catchments. A comparison of the corrected datasets using contour maps shows that the proposed modelling scheme can reliably reduce the systematic bias at a grid

point that is not used in the process of parameter estimation. In particular, the contour map with the 99[th] percentile shows a more accurate representation of the observed daily rainfall than other combinations. The findings in this study suggest that the proposed approach can provide a useful alternative to readers who consider the bias correction of a regional-scale modelled data with a limited network of rain gauges.

Although the study has been carried out in South Korea, the methodology has its potential to be applied in other parts of the world.

## 1. Introduction

Recent studies have documented that there are long-term climate effects on a wide range of fields such

as agriculture, environment, health, economy and water resources (IPCC, 2014; Nelson et al., 2009; Patz et al., 2005; Vörösmarty et al., 2000). Of these fields, water related hazards such as floods and droughts are one of the main concerns for water resource managers.

To systematically assess water resources and water related hazards, it is necessary to collect reliable long-term climate data. Locally recorded data have played an important role, and they have been

considered to be accurate values in the modelling process. However, it has been widely acknowledged that the observed data are often coarsely represented in the model calibration, and long-term climate data are not readily available in many countries around the world. For instance, South Korea has an area of about 100,032 km$^2$, but only a few dozen stations have continuous records of daily time scales over the past 40 years, and the stations with long records over 50 years are still fewer than 20. For these reasons,

reanalysis datasets based on modern data assimilation techniques have been produced and used to explore the global-, continental- and country-scale climate change (Dee et al., 2011; Donat et al., 2016; Gao et al., 2016; Hersbach et al., 2015; Poli et al., 2016; Zhang et al., 2013). A primary strength of the reanalysis data is that they provide information on a spatially finer scale with a longer period, a few of which can



cover the whole 20[th] century. For example, the National Oceanic and Atmospheric Administration (NOAA) has produced the 20[th] century reanalysis (20cR) which spans from 1850 to 2014, and the European Centre for Medium-Range Weather Forecasts (ECMWF) has also released century-long datasets such as the ECMWF 20[th] century atmospheric model ensemble (ERA-20cm) and ECWMF 20[th] century assimilation surface observations only (ERA-20c), which cover years from 1900 to 2010 (Compo et al., 2011; Hersbach et al., 2015; Poli et al., 2016). All of them can globally provide daily or sub-daily scale precipitation data, but differences exist in the assimilation techniques and spatial-temporal resolution. The products from the ECMWF (such as ERA-20c and ERA-20cm), are based on the Integrated Forecasting System version Cy38 r1 with 0.125° spatial resolution, which are more relevant in regional-scale studies in South Korea due to their higher spatial resolution. The difference between ERA-20c and ERA-20cm is that the former assimilates pressure and wind observations but the latter does not consider them in the modelling process (Donat et al., 2016; Hersbach et al., 2015; Poli et al., 2016). Therefore, ERA-20cm is limited in reproducing the actual synoptic situation (Gao et al., 2016; Hersbach et al., 2015). On the other hand, NOAA-20cR was processed by an Ensemble Kalman Filter technique (Compo et al., 2011), but its spatial resolution (i.e. 1.875°×1.9°) is much coarser than the other century-long reanalysis data. Under these conditions, this study has selected the ERA-20c daily precipitation data with 0.125°×0.125° spatial resolution, as an alternative for the observation in climate impact assessment over South Korea.

However, although substantial improvement have been made in the modeling process, previous studies have shown that reanalysis datasets still have their own systematic errors which vary in space and time (Bao and Zhang, 2013; Bosilovich et al., 2008; Gao et al., 2016; Ma et al., 2009). Hence, to effectively



reduce the uncertainty of the global reanalysis, it is important to identify the reasons for the biases and

apply a relevant bias correction method before applying the data in hydrological modeling. However,

there are limited studies on bias correction for ERA-20c daily precipitation in hydrologic applications.

Most of the existing studies have been performed mainly within the context of comparison across different

reanalysis data, but not bias correction issues (Donat et al., 2016; Poli et al., 2016). Thus, to better

understand the biases and their roles in hydrologic applications, this study focuses on exploring bias

correction methods, especially for extreme value analysis.

The underlying concepts for the bias correction approach vary from a simple linear regression to a

sophisticated distribution mapping approach (Teutschbein and Seibert, 2012). There are four distinct types

of representative bias correction methods: linear scaling, local intensity, power transformation, and

quantile mapping (QM) (Fang et al., 2015; Schmidli et al., 2006; Teutschbein and Seibert, 2012).

Although each method has its own merits and limitations, previous studies have shown that bias correction

methods were generally capable of reducing systematic errors in numerical models and, among them, QM

showed better performance than other approaches, especially for precipitation (Fang et al., 2015; Jakob

Themeßl et al., 2011; Teutschbein and Seibert, 2012). The QM method, referred to as other names such

as 'distribution mapping' and 'probability mapping', was used to rectify the cumulative distribution of

the modelled data against that of the observed data by employing a transfer function, which is usually

based on a gamma distribution for the daily precipitation.

However, there are two main drawbacks to the QM approach based on a gamma distribution (gQM).

First, it has been acknowledged that gQM often fails to reproduce extreme rainfall, which is mainly

described by the upper tail of the distribution (Hundecha et al., 2009; Volosciuk et al., 2017; Vrac and



Naveau, 2007; Wilks, 1999). In other words, the gQM approach results in underestimation of the extreme rainfalls, which, in turn, leads to underestimation of the design rainfalls. On the one hand, one may intuitively consider the heavy tailed distributions such as extreme value distribution (e.g. Gumbel distribution, generalized extreme value distribution and Weibull distribution). On the other hand, the heavy tailed distribution for the bias correction may result in overestimation of daily rainfall in the lower tail of the distribution. In these contexts, a composite distribution including the mixture distribution (such as the Pareto mixture distribution) has been applied to the quantile mapping approach, especially for the correction of climate change scenarios (Gutjahr and Heinemann, 2013; Nyunt et al., 2016; Smith et al., 2014; Volosciuk et al., 2017). Comparatively little attention has been given to the bias correction of the reanalysis data. In these contexts, this study aims to introduce a quantile mapping approach based on a composite distribution of a generalized Pareto distribution (GPD) for the upper tail (e.g. 95th and 99th percentile) and a gamma distribution for the interior part of the distribution.

The conventional QM method is also limited in that it cannot be applied directly to the ungauged basin, where a one-to-one mapping between the observed and the modelled data does not exist. More specifically, only a transfer function of a set of grid points for the paired precipitation data can be obtained. Thus, an alternative method for the synthesis of unpaired data needs to be established. The general approaches to the interpolation of in-situ data for the quantile mapping are the inverse distance weighting (IDW) and the kriging method, and the interpolated values can then be used to obtain the transfer function for the ungauged basin. For example, Gutjahr and Heinemann (2013) applied the IDW method to produce spatially continuous estimates of the daily precipitation for the spatial bias correction. However, the systematic error in the process of the spatial interpolation of daily rainfall can be propagated through to



the parameter estimation in the quantile mapping approach. Thus, a primary question in the statistical bias correction analysis is whether the QM method can reliably improve ERA-20c daily precipitation over 100 years when including the ungauged sites.

From this background, this study mainly focuses on exploring the following questions:

*(1) What are the characteristics of the uncertainty associated with the ERA-20c daily precipitation data in South Korea? Do the reanalysis data well describe the statistical properties in terms of the extreme as well as the mean values?*

*(2) How well does the traditional QM method approach perform on the reanalysis data? Can a combined distribution based bias correction be more effective for the reduction of the systematic*

*error compared with the bias correction approach based on a single distribution (gQM)?*

*(3) How can we effectively extend the combined distribution approach to the spatial bias correction for ungauged catchments? Can the proposed scheme facilitate a reconstruction of long-term precipitation, especially for the estimation of annual maximum series (AMS) of daily precipitation?*

To address these questions, we investigated the bias correction in three phases. First, we attempted to

understand the statistical behavior of the ERA-20c data and further analyze the biases and errors in the reanalysis mean and extreme precipitation. Second, the QM approach was explored by using a combined Gamma-Pareto distribution in the bias correction method to better represent the upper tail of the distribution for 48 stations for the baseline period 1973-2010. The corrected data for the proposed approach were then compared with that of the observed. Finally, we proposed a spatial bias correction

approach based on the parameter contour maps (IM-PCM). The correction approach consists of three steps for ungauged catchments. The reanalysis data and observed precipitation are summarized in Section

2. The theoretical background for the proposed bias correction approach is introduced in Section 3. The proposed model was applied to the daily rainfall data for the baseline period and a retrospective analysis of the data was then conducted for the estimation of AMS rainfalls in Section 4. Finally, concluding remarks are provided in Section 5.

## 2. Study area and data

### 2.1 Study area and local gauged data

South Korea is located in the northeast part of Asia, and lies between latitudes 33°-39°N and longitudes 125°-132°E, including all the islands. The total area is approximately 100,032 km$^2$, and its annual average

rainfall is about 1,277 mm. In South Korea, there are hundreds of local weather stations available. However, most of them have been installed after 1970, and only a few stations provide long-term daily precipitation records for more than 40 years. In this study, 48 local rain gauges, spanning from 1973 to 2010, are used for the bias correction and its evaluation over South Korea. The daily precipitation sequences for the reference period (1973-2010) were obtained and compiled from the Korea

Meteorological Administration (KMA). The location of the study area and the local gauging stations used in this study are illustrated in Figure 1, and the details for the stations are summarized in Table 1.

**[Insert Figure 1 and Table 1]**

### 2.2 ERA-20c daily precipitation

As previously mentioned in Section 1, we explored the ERA-20c daily precipitation, which is one of the longest reanalysis data covering the whole 20$^{\text{th}}$ century (Donat et al., 2016; Poli et al., 2016). The

ERA reanalysis system is based on a set of data assimilation schemes, and the system provides relatively high resolution gridded datasets, including daily total precipitation from 1900 to 2010 via the ECMWF web server. In this research, we focused on the data from the mainland of South Korea from January 1973 to December 2010 with its highest resolution, 0.125°×0.125° (approximately 13.8 km×11.2 km), which

consists of 603 grid points. The data taken over the sea were excluded from this study. The specific gridded points for ERA-20c are illustrated in Figure 1.

It is crucial to understand the features of the model biases to improve the modelled reanalysis data. Some of the general features of ERA-20c daily precipitation over South Korea are examined in terms of the mean and the extreme values. For the mean precipitation, we compared the intra-seasonal variability

within the annual cycle by exploring the monthly means and the 10-day running means between the observed and ERA-20c precipitation (as shown in Figure 2) averaged over all 48 stations during the baseline period (1973-2010). The model performance was evaluated by both the Nash-Sutcliffe efficiency (NSE) and root-mean-square error (RMSE), which are described in Section 3.4. The results confirmed that ERA-20c can reproduce the mean values quite well, while there is a significant difference between

modelled and observed precipitation during the summer season (i.e., July to September), which may lead to an underestimation of extreme rainfall.

**[Insert Figure 2]**

In terms of the extreme rainfall episodes, the 50 top events were extracted for the baseline period, and an underestimation of extremes in the ERA-20c was clearly identified, as illustrated in Figure 3. The

deviations are generally large, even for relatively larger upper tail parts of the distribution with -1.088 for NSE and 76.69 mm for RMSE (Figure 3(a)). On the one hand, the deviations are quite systematic in the

sense of the bias correction. The relationships between the 50 top extreme rainfalls in the stations 4, 16, 28 and 40 show that the discrepancies were largely attributed to differences in rainfall during summer season, as noted in Figure 2. The bias in extreme values is proportional to the amount of rainfall, and the biases are likely to be higher in the upper tails of the distribution than that of the middle layer, as shown

in Figure 3(b).

**[Insert Figure 3]**

In summary, the ERA-20c precipitation data are capable of reliably reproducing the mean values, while the extreme values are consistently underestimated. The results obtained here could indicate that although the climate models adequately represent the mean climate of the historical period, heavy rainfalls in the

summer season can be significantly underestimated due to fact that intensive rainfall events driven by convective storms may not be effectively resolved by the current climate modelling approach and spatial resolution. On the other hand, as shown in Figure 4, a much higher frequency of wet-days was observed for all months. More generally, the over-pronounced frequency of light precipitation by climate models is a well-known problem, and it may partially cause the underestimation of the extremes. In these contexts,

a two-stage bias correction approach to daily precipitation is typically adopted to first adjust the overestimated wet-day frequency and then rectify the biases associated with both the mean and extreme values.

**[Insert Figure 4]**

**3. Methodology**

As illustrated in the previous section, two deficiencies in the ERA-20c became evident: the



overestimation of the wet-day frequency and underestimation of the extreme values. To correct the biases, we adopted a two-stage bias correction scheme that consists of the wet-day frequency correction scheme and the composite distribution based QM approach. The proposed methods and their assumptions used in this study are provided in this section.

### 3.1 Wet-day frequency correction scheme

It is well known that the wet-day frequencies of the simulated precipitation data from climate models are typically inflated due to the generation of small precipitation amounts near 0.1 mm/day (Kim et al., 2015b; Nyunt et al., 2016; Piani et al., 2010). For this reason, a cut-off threshold (TH) approach has been

commonly applied to adjust the wet-day frequency in the bias correction for daily precipitation using different criteria (Jakob Themeßl et al., 2011; Kim et al., 2015a, 2015b; Nyunt et al., 2016; Piani et al., 2010; Rabiei and Haberlandt, 2015; Schmidli et al., 2006; Volosciuk et al., 2017). For example, Piani *et al.*(2010) and Volosciuk *et al.* (2017) applied 0.1 mm/day as the threshold, whereas the wet-day frequency of simulated precipitation was set equal to that of the observed (Kim et al., 2015a, 2015b; Nyunt et al.,

2016). Rabiei and Haberlandt (2015) compared five different thresholds (0 mm/hr, 0.02 mm/h, 0.05 mm/h, 0.07 mm/h, 0.1 mm/h) for spatial bias correction of hourly radar data and concluded that the threshold 0.05 mm/h performed the best among the five in terms of the reduction of biases.

In our study, a set of predetermined thresholds were used to adjust the wet-day frequency of the modelled daily precipitation from ERA-20c. We considered four different thresholds to identify an

20 optimal threshold (TH) for the ERA-20c: (TH1) 0>mm/day, (TH2) 0.1>mm/day, (TH3) 1>mm/day, and (TH4). The frequency of wet days was set to the observed value. On the other hand, changes in the wet-





day frequency can affect the overall performance in the bias correction process through the QM approach,

because a transfer function between the simulated and observed precipitation is established on the basis

of non-zero precipitation. In this context, the optimum threshold was evaluated through the experiment

with gQM for a pair of daily rainfall series for each station. It should be noted that daily rainfalls below

the thresholds were set to zero for ERA-20c. Among four thresholds, the determined threshold was then

applied in the next steps.

**3.2 Statistical Bias Correction Model: QM with a composite distribution**

A main concept of QM is to map the modelled data to the observed data in the probability space. More

generally, cumulative distribution functions (CDFs) of the modelled data are mapped to that of the

observed, which is considered "true" (Rabiei and Haberlandt, 2015; Teutschbein and Seibert, 2012). In

other words, the distribution of simulated values is fitted to the true distribution, the relationship of which

is established in the advanced stages of bias correction. A gamma distribution with two parameters has

been commonly used in the previous studies since it can describe the main features of daily precipitation

(Kim et al., 2015a, 2015b; Piani et al., 2010; Teutschbein and Seibert, 2012). The gamma distribution and

its transfer function for the QM can be expressed as follows:

$$F(x|\alpha,\beta) = \frac{1}{\beta^{\alpha}\Gamma(\alpha)} \int_0^x t^{\alpha-1} e^{-t/\beta} dt \; ; \; x \geq 0; \; \alpha,\beta > 0 \qquad (1)$$

$$x_{cor} = F^{-1}[F(x_{mod}; \alpha_{mod}, \beta_{mod}); \alpha_{obs}, \beta_{obs}] \qquad (2)$$

where, $x_{cor}$ and $x_{mod}$ are the corrected data and the uncorrected (or modelled) data in the baseline

period. $F$ is a gamma CDF and $F^{-1}$ is its inverse function, while $\alpha$ and $\beta$ are the shape and scale



parameters of the gamma distribution, respectively. To account for the seasonality, it is common to have

bias correction models for each month that are independent from the others (Kim et al., 2015b).

To effectively improve the bias in the extreme rainfall for ERA-20c, we propose a composite

distribution based on the QM approach which is comprised of different types of distributions. More

specifically, the extreme value distribution can be utilized for the upper tail of the distribution, while a

gamma distribution is applied for the interior part of the distribution. For extremes, the 95[th] or 99[th]

percentiles have been applied as an upper threshold in numerous studies because the distribution of

excesses over the high thresholds is asymptotically approximated by a generalized Pareto distribution

(GPD) (Acero et al., 2011; Chan et al., 2015; Gutjahr and Heinemann, 2013; Manton et al., 2001; Nyunt

et al., 2016; Wilson and Toumi, 2005). In this study, we apply both the 95[th] and 99[th] percentiles as the

upper thresholds.

The GPD has been widely applied to the peak-over-threshold (POT) series for the selection of the best-

fit distribution for the extreme rainfalls (Gutjahr and Heinemann, 2013; Hundecha et al., 2009; Nyunt et

al., 2016; Volosciuk et al., 2017; Vrac and Naveau, 2007), although there have been a considerable number

of studies using other extreme value distributions including: the generalized extreme value (GEV),

Weibull (WEI), Gumbel (GUM), and Log-normal (LOGN). To ensure the suitability of the GPD, we first

evaluated six different distributions, GPD, GEV, GUM, WEI, LOGN and gamma, for the extremes in

both the observed and ERA-20c over the 95[th] and 99[th] percentiles using the Akaike Information Criterion

(AIC) and Bayesian Information Criterion (BIC). For a given threshold, the GPD was selected as the best-

fit distribution for the extremes as shown in Table 2.





**[Insert Table 2]**

As previously mentioned, the GPD is separately applied to the extreme values defined by the 95th and 99th thresholds at each station as a transfer function, whereas the gamma distribution was mainly applied to the interior part of the distribution. Again, note that we assume that the GPD is used for the upper tail of the distribution while the gamma is used for the remainder, as illustrated in Equation (3).

$$x_{cor} = \begin{cases} F^{-1}_{obs,gamma}(F_{mod,gamma}), & \text{if } x \leq 95 \text{ th or } 99 \text{ th percentile} \\ F^{-1}_{obs,GPD}(F_{mod,GPD}), & \text{if } x > 95 \text{ th or } 99 \text{ th percentile} \end{cases} \tag{3}$$

Here, $F_{mod,gamma}$ and $F_{mod,GPD}$ are the CDFs of the ERA-20c model for gamma and GPD. Similarly, $F^{-1}_{obs,gamma}$ and $F^{-1}_{obs,GPD}$ are the inverse (or quantile) function of CDFs of observations for gamma and GPD, respectively. The heavy tailed distribution for POTs is defined as follows for a GPD with a high upper threshold ($u$) (Coles, 2001; Gutjahr and Heinemann, 2013):

$$F(x) = P_r(X - u \leq x \,|\, X > u) = \begin{cases} 1 - \left(1 + \dfrac{\xi x}{\theta}\right)^{-\frac{1}{\xi}} & for\ \xi \neq 0 \\ 1 - \exp\left(-\dfrac{x}{\theta}\right) & for\ \xi = 0 \end{cases} \tag{4}$$

Here, $\theta = \sigma + \xi(u - \mu)$ is the reparametrized scale parameter, and $\xi$ is the shape parameter. In this study, the thresholds ($u$, the 95th or 99th percentile) for observed and modelled precipitation were derived at each station.

In this approach, the four parameters to be estimated are the shape ($\alpha$) and scale ($\beta$) parameters for the gamma distribution, and the shape ($\xi$) and scale ($\theta$) parameter for GPD, while the upper thresholds are assumed to be known for the given 95th or 99th percentile. The parameters for gamma distribution are



estimated on a monthly basis, whereas the parameters of GPD are estimated using entire POTs for all months in each station. Here, the maximum likelihood method is mainly used to estimate all the parameters. Hereafter, the proposed method with a composite distribution of gamma and GPD is referred to as gpQM. Moreover, the gpQM with the 95[th] and 99[th] upper thresholds were abbreviated as gpQM95

and gpQM99, respectively. For comparison, the conventional bias correction gQM was also applied and compared in terms of the accuracy of both the extreme and the mean value.

### 3.3 Spatial interpolation by parameter contour maps

In the gpQM approach, a pair of observed and modelled data are required to estimate the six parameters

(TH, $\alpha$, $\beta$, $\theta$, $\xi$ and $u$). However, because there is a limited number of available weather stations, the transfer function for the QM could not be established for all grid points. Therefore, the existing methods can only be applied over gauged catchments. In contrast, we introduce an interpolation method based on parameter contour maps (IM-PCM) which consist of three steps as summarized in Figure 5. For gpQM95 and gpQM99, the six parameters (TH, $\alpha$, $\beta$, $\theta$, $\xi$ and $u$) were first estimated for each station as already

noted in Sections 3.1 and 3.2. Secondly, a contour map for each parameter was then constructed using a 2-dimensional linear interpolation technique as shown in Figure 6. Finally, a set of parameters for the gpQM were taken from the maps to construct the transfer function for all grid points. The cut-off threshold (TH) is the first interpolated variable, and the maps of shape ($\alpha$) and scale ($\beta$) parameters for the gamma distribution were then generated on a monthly basis, while the shape ($\theta$), scale ($\xi$) and upper threshold

($u$) parameter maps of the GPD were created by using the entire POTs on an annual basis. For the gQM, a similar process to the one described above was used to produce three parameter (TH, $\alpha$ and $\beta$) maps





for the transfer function.

**[Insert Figures 5 and 6]**

### 3.4 Evaluation criteria

A main goal of this study is to evaluate the suitability of the bias-corrected ERA-20c in terms of both the extreme and the mean values. For the extremes, we compared the rainfalls for a given 99[th] threshold between three different QM approaches including gQM, gpQM95 and gpQM99. In addition, the annual maximum series (AMS) for all stations were extracted and compared to that of the corrected ERA-20c. For the mean values, both the monthly mean and 10-day running means between the observed and ERA-20c precipitation were compared in the context of the intra-seasonal variability. Moreover, we used the root mean square error (RMSE), and Nash-Sutcliffe efficiency (NSE), which are well known goodness-of-fit measures for model evaluation in the field of hydrology (Legates and McCabe Jr., 1999). These are provided in Equations 5 and 6:

$$RMSE = \sqrt{\frac{\sum_{i=1}^{n}\left(Y_i^{obs} - Y_i^{sim}\right)^2}{n}} \tag{5}$$

$$NSE = 1 - \left[\frac{\sum_{i=1}^{n}\left(Y_i^{obs} - Y_i^{sim}\right)^2}{\sum_{i=1}^{n}\left(Y_i^{obs} - Y_i^{mean}\right)^2}\right] \tag{6}$$

Here, $Y_i^{obs}$ is the $i$-th observation, $Y_i^{mean}$ is the mean of the observation, while $Y_i^{sim}$ is the modelled data, and $n$ is the number of observations. For the NSE, the dataset accuracy improves as the efficiency approaches 1.

The performance of the proposed interpolation method was evaluated by a leave-one-out procedure

within a cross validation framework. To be more specific, this approach estimates a set of parameters for the observation of daily precipitation for 47 stations out of 48 stations, and the estimated parameters were further used to build contour maps as shown in Figure 6. The set of parameters of the grid point corresponding to the excluded station were taken from the maps, and the proposed bias correction approaches were then applied. Again, note that the model performance for the extreme and mean values were evaluated with regard to RMSE and NSE as described in Section 3.4.1.

## 4. Results and Discussion

### 4.1 Evaluation for the lower threshold

This study examined four different thresholds (TH1, TH2, TH3, and TH4) for adjustment of the wet-day frequency of ERA-20c daily precipitation through an experiment with the QM approach in terms of both the mean and extreme values. We investigated the intra-seasonal variability within the annual cycle by comparing the monthly means and the 10-day running means as an overall evaluation of the bias corrected precipitation. Here, all the values were averaged over all 48 stations during the baseline period (1973-2010) as illustrated in Figure 7. We found that the threshold TH4 yielded the best results among the four in terms of the reduction of biases, as summarized in Figure 7(a) and Table 3. Again note that TH4 is the case where the frequency of wet days of ERA-20c is set to that of the observed. On the other hand, the other thresholds, TH1, TH2 and TH3, showed a significant overestimation, whereas the uncorrected ERA-20c showed a relatively small bias. Our results offer insight on how improper thresholds for the wet-day frequency may affect bias correction results, leading to a significant overestimation of daily rainfall. Such discrepancies may arise from the significantly different thresholds used to adjust the wet-day frequency.

As illustrated in the previous section, the lower thresholds for TH4 were varied over the range 0-4.66 mm while the thresholds assumed in the TH1, TH2 and TH3 are much lower than the one measured in the TH4, especially for the summer season (July-September). Indeed, the similar results seen in the 10-day moving mean suggests that our findings may be generalizable to cut-off thresholds seen in different

locations and seasons, as shown in Figure 7(b) and Table 3. We also found that the degree of bias associated with the cut-off thresholds significantly varied within a specific season, especially in the summer.

**[Insert Figure 7 and Table 3]**

For the evaluation of the extreme rainfalls associated with different thresholds, we extracted rainfall

events exceeding a given 99[th] threshold and we compared the four different thresholds for all stations. As illustrated in Figure 8, a systematic significant underestimation of extremes in the ERA-20c is most apparent, while the improvements appear to result from enhanced representation of the bias associated with extreme values regardless of the threshold. Specifically, TH4 performs the best with 0.755 for NSE and 27.33 mm for RMSE, followed by TH3, TH2 and TH1. Given these results, TH4 could be the most

reliable cut-off threshold for the ERA-20c under the gQM approach. On the other hand, there remains considerable potential for improving extremes, especially over 300 mm/day. Thus, we will further explore the bias correction approach for the upper tail of the distribution.

**[Insert Figure 8]**

**4.2 Bias correction based on a composite Gamma-GPD distribution**

This study introduces a composite (or piecewise) distribution based QM approach which consists of





gamma distribution and GPD, for a given set of thresholds. Here, the 95[th] or 99[th] quantiles have been considered as an upper threshold for the correction of extremes (gpQM95 and gpQM99). The composite distribution approach was evaluated by comparing the obtained extreme rainfalls from modelled ERA-20c with the ones observed for the baseline, as shown in Figure 9. In comparison with the extreme daily rainfalls over the 99[th] percentile, the GPD based bias correction schemes (i.e., gpQM99 and gpQM95) demonstrate better performance in terms of reproducing the extremes than gQM (Figure 9(a)). gpQM99 shows the best performance in terms of NSE with an efficiency of 0.906, and a good agreement was achieved with 0.879 in gpQM95, whereas the gQM was 0.755. For RMSE, gpQM99 (i.e., 16.92 mm) and gpQM95 (i.e., 19.16 mm) showed a significant reduction of the errors by 38.1% and 29.9% relative to gQM (27.33 mm). Moreover, a comparison of the AMS rainfall also confirmed that gpQM99 and gpQM95 were capable of reproducing rainfall characteristics observed in the AMS more effectively than gQM. Specifically, gpQM99 showed the best performance with 0.912 for NSE and 18.80 mm for RMSE, whereas gpQM95 was 0.892 for NSE and 20.77 mm for RMSE. The results obtained in this study suggest that the gpQM approach is more appropriate to reduce the systematic errors in estimating extreme rainfalls than gQM.

**[Insert Figure 9]**

Apart from evaluating the models in the extreme cases, it is important to ensure that the proposed bias correction model with the GPD can reproduce the mean values as well. Again, we evaluate both the monthly mean and 10-day moving mean of the corrected daily precipitation as shown in Figure 10 and Table 4. For the monthly mean, gQM and gpQM99 give the best performance (Figure 10(a)), leading to the highest efficiency for NSE of 0.997 for both methods, and the lowest RMSE, about 4.77 to 5.12

mm/month, respectively (Table 4). For gpQM95, the efficiency for NSE is close to one, but the RMSE, 9.41 mm/month, is nearly twice those of gQM and gpQM99. In terms of the 10-day moving mean, the results have shown that all QM approaches work equally well, although gpQM99 offers the best performance (Table 4). More generally, the gpQM99 approach can effectively correct the biases associated with the upper tails of the distribution without a loss in the efficiency of the bias correction process.

**[Insert Figure 10 and Table 4]**

It should be noted that the bias still remains large in the summer season as seen in the 10-day moving mean. The difference was mainly attributed to the discrepancies in the seasonal or monthly distribution of the heavy rainfall events between the observed and modelled data (Nyunt et al., 2016). In other words, there is a clear difference in the monthly number of extreme events over the 95[th] or 99[th] thresholds between the observed and ERA-20c (Figure 11), and this is considered to be the main source of the bias in terms of extremes in the intra-seasonal band. The results obtained in these experiments imply that the upper thresholds could be different (or updated) for each month to better represent the intra-seasonal change. On the other hand, estimation of different thresholds on the monthly basis could lead to unreliable estimates of extreme values due to insufficient data for estimating the GPD parameters.

**[Insert Figures 11]**

### 4.3 Spatial interpolation on bias correction parameters

The proposed IM-PCM approach is validated by leave-one-out cross validation. In this study, we estimated a set of parameters for the observation of daily precipitation, and the estimated parameters were

then used to build contour maps. For extreme values of the interpolated daily precipitation, POTs exceeding a given 99$^{th}$ percentile and AMS were first constructed and compared between three different QM approaches including gQM, gpQM95 and gpQM99. Note again that all results were obtained from the cross-validation procedure having considered different possible samples. As illustrated in Figure 12 (a), the corrected extremes using an interpolated set of parameters by IM-PCM showed good agreement with the observed values for the three QMs. Among them, gpQM95 and gpQM99 gave the best performance for the given POTs (Figure 12 (a)) with 0.781 for NSE, and 0.741 for gQM. Similar results were obtained for the RMSE. Moreover, the proposed gpQM99 approach using the interpolated parameters was capable of reproducing the AMS with 26.35 mm for RMSE and 0.827 for NSE (Figure 12 (b)). However, it should be noted that an increased bias exists, which is largely attributable to the parameter interpolation process. For example, the RMSE in AMS using gpQM99 with IM-PCM increased from 18.80 to 26.35 mm for RMSE when compared with a pointwise bias correction as already seen in Figure 9(b). A similar increase (i.e. 20.77 to 26.30 mm) was also observed in the gpQM95. Nevertheless, the RMSE for the corrected AMS data by IM-PCM with gpQM99, 26.35 mm, is still smaller than that of the pointwise bias correction from gQM, 28.07 mm.

**[Insert Figure 12]**

In terms of the mean precipitation, the monthly mean and 10-day moving average of bias corrected rainfall using a set of parameters obtained from IM-PCM were evaluated (Figure 13 and Table 5). Although all three QM approaches yielded slightly different estimates, overall favorable performance was obtained for the monthly mean with a model efficiency over 0.98 for NSE. Among the options, gQM and gpQM99 performed the best and showed the lowest RMSE (Figure 13(a) and Table 5). Figure 13 (b) shows a similar

result for the 10-day moving average with an efficiency over 0.96 for NSE. Given these results, the

proposed gpQM99 approach with IM-PCM can effectively rectify the spatial-temporal bias of the ERA-

20c model data without a loss in efficiency for the mean values.

**[Insert Figure 13 and Table 5]**

It is well known that precipitation is mainly influenced by the topology in mountainous areas, so

numerous studies have used elevation as an exogenous factor for rainfall interpolation (Adhikary et al.,

2017; Goovaerts, 2000; Lloyd, 2005). We therefore explored the relationship between the elevation and

parameters for all 48 stations. As summarized in Table 6, the Pearson correlation $r$-values were not

statistically significant, leading to a weak dependence between the elevation and parameters. The results

imply that the elevation may not be important in terms of the interpolation of the parameter. In summary,

the proposed interpolation scheme for the QM approach provided bias corrected long-term precipitation

data, especially for ungauged catchments. On the other hand, the proposed approach was easy to use and

may help to reduce bias associated with the interpolation of daily precipitation. Moreover, this approach

can be further used to obtain a century-long daily precipitation series over the Korean peninsula, which

could be useful in terms of reducing uncertainty in the parameter estimation of rainfall frequency analysis.

**[Insert Table 6]**

The bias correction methods developed in this study both improved the quality of the data and extended

daily precipitation over the 20[th] century in South Korea. More specifically, this study further utilizes the

derived transfer function for the baseline period 1973-2010 to provide the daily precipitation for the period

1900-2010 under the stationary assumption. Finally, we explored changes in the mean and extreme using

the gpQM99 approach for three different periods, 1900-1972, 1973-2010 and 1900-2010, in the context

of a retrospective analysis. As shown in Figure 14 (a), the evaluation results for the monthly mean show a very noticeable and sudden increase in the recent period, especially for the summer season (July-September), while no significant changes were observed for dry season (October-April). Figure 14 (b) shows boxplots representing a distribution of the AMS for the three periods. The distribution of the AMS

derived from the gpQM99 approach for the period 1973-2010 was almost identical to that of the observed, which indicates that the proposed gpQM99 was capable of reproducing the extremes of daily precipitations. As expected from the changes in summer rainfall, the distribution of the AMS for the recent period 1973-2010 is much wider than that of the period 1900-1972 (i.e. gpQM99-1), especially for the upper tail of the distribution. This may lead to an increase in design rainfalls for a specific return

period. On the other hand, the distribution of the AMS for the entire period 1900-2010 is quite similar to that of the observed in terms of median AMS, while its range is relatively narrower than the recent period.

**[Insert Figure 14]**

## 5. Concluding remarks

The main objective of this study was to explore the century-long reanalysis data, ERA-20c, especially for daily precipitation over South Korea in the context of bias correction. We first investigated the utility of the ERA-20c data as a proxy data over South Korea for hydrological applications and further examined several issues concerning the aspects of the bias correction that influence the use of modelled data in practice. In general, we found that there is a fairy good agreement between the observed and the ERA

reanalysis data for the baseline period 1973-2010. On the one hand, the results obtained here have shown that the ERA-20c precipitation data still have their own systematic biases, particularly in the frequency

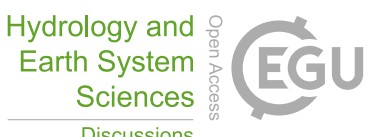

of wet-days and the extreme upper tail of the distribution. More specifically, the over-pronounced frequency of wet-days and the considerable underestimation of daily precipitation have been identified in the ERA-20c over South Korea. Given these results, we proposed a two-stage bias correction approach to daily precipitation, which is comprised of two distinct parts: a model for adjusting the overestimated wet-day frequency and a model for reducing the biases associated with extreme values. To adjust the wet-day frequency, we explored four different thresholds through an experiment with the QM approach. In terms of extremes, a composite Gamma-GPD distribution based QM approach was introduced. Finally, we proposed an IM-PCM approach as an alternative to constructing the transfer function for the ungauged basin. The key findings obtained in this analysis are summarized as follows:

1. Our findings are consistent with the notion that the mean daily precipitation is reproduced well by the reanalysis. Our study also confirms that the mean and annual cycle of daily precipitation as observed over South Korea is well simulated by the ERA-20c reanalysis. However, considerable underestimation of the daily maximum precipitation was consistently seen in the ERA-20c, especially during the summer season. The results presented here illustrate that the heavy rainfalls in the summer season could be significantly underestimated by the current climate modelling system, although the reanalysis system adequately reproduces the mean climate of the historical period. Another issue with respect to the evaluation of ERA-20c daily precipitation is related to the much higher frequency of wet-days than that of the observed, which may in turn influence the underestimation of the extremes.

2. In this study, a two-stage bias correction approach to the ERA-20c precipitation was proposed to



adjust the overestimated wet-day frequency and the biases associated with the upper tail of the distribution. In terms of the wet-day frequency, we examined four different types of thresholds (i.e., TH1, TH2, TH3 and TH4) to identify an optimal threshold. TH4 is the case where the frequency of wet-days of ERA-20c is set to that of the observed and produces the best results among the four. Moreover, TH4 is allowed to have different thresholds for each month, unlike the other three approaches (i.e., TH1, TH2 and TH3) in which a fixed value was assumed over all the months for all the stations. Our results offer insights on how inappropriate thresholds for the wet-day frequency may significantly influence the bias correction results. To better represent the bias in the extreme rainfall, we proposed a composite distribution based QM approach, which consists of the gamma distribution and GPD for the two thresholds (i.e., the $95^{th}$ and $99^{th}$ percentiles). Given the efficiency gains, this study suggests that the gpQM approach is more appropriate to reduce the systematic errors in estimating extreme rainfalls than gQM. To be more specific, the gpQM99 approach can effectively reduce the biases in the upper tails of the distribution without a loss of efficiency in the overall bias correction process. However, a large bias still exists in the summer season, and thus the bias in extreme rainfall that the qpQM99 offers in the process of bias correction suggests that the ERA-20c data might be insufficient in terms of reflecting the specific regional patterns associated with extreme rainfall over South Korea.

3. We explored an alternative to obtain the transfer function of the QM approach for the ungauged catchments in the context of the cross-validation process. From this perspective, we have proposed an interpolation method based on parameter contour maps (IM-PCM), which is based on the interpolation of the five parameters over the entire region of interest. The corrected daily

precipitation series using an interpolated set of parameters by the IM-PCM showed good agreement with the observed precipitation, and particularly the proposed gpQM99 with the IM-PCM performs the best in terms of reducing the spatial-temporal bias of the ERA-20c model data without a loss of efficiency. We finally utilized the derived transfer function for the baseline period 1973-2010 to

extend the daily precipitation for the period 1900-2010 under the stationary assumption, and we examined the changes in daily precipitation for three different periods, 1900-1972, 1973-2010 and 1900-2010, as a retrospective analysis. We found that a very noticeable and sudden increase in the recent period was observed during the summer season (July-September).

The findings demonstrated in this study help to understand the knowledge gaps about the bias correction of the century-long reanalysis, ERA-20c, as well as the key characteristics of daily precipitation over South Korea. Further, the results obtained here can provide a useful perspective on the bias correction of the modelled data in the reanalysis and regional climate modelling systems for the regional-scale analysis with a limited network of rainfall stations. The impact of climate change on water resources using the

extended daily precipitation data for the period 1900-2010 will be explored further. Although the study has been carried out in South Korea, the methodology has the potential to be applied in other parts of the world. We hope this paper will stimulate the hydrometeorological community to explore the issues raised in the long-term reanalysis data in other countries under different climate and geographical conditions.



**Acknowledgements**

The first author is funded by the Government of South Korea for carrying out his doctoral studies at the University of Bristol. We are grateful for the relevant data provided by KMA and ECMWF. The second author is supported by a grant (17AWMP-B121100-02) from Advanced Water Management Research

5   Program (AWMP) funded by Ministry of Land, Infrastructure and Transport of Korean government.





## Appendix A. List of Abbreviations

| ID | Definitions |
|---|---|
| AIC | Akaike information criterion |
| AMS | Annual maximum series |
| BIC | Bayesian information criterion |
| CDF | Cumulative distribution functions |
| ECMWF | European Centre for Medium-Range Weather Forecasts |
| ERA-20c | ECWMF's 20th century reanalysis assimilated by surface observations only |
| ERA-20cm | ECMWF's 20th century atmospheric model ensemble |
| GEV | Generalized extreme value distribution |
| GPD | Generalized Pareto distribution |
| gpQM | Quantile mapping approach based on a composite distribution of gamma and GPD |
| gpQM95/ gpQM99 | gpQM with the upper tail of 95th/99th percentile |
| gQM | Quantile mapping approach based on a gamma distribution |
| GUM | Gumbel distribution |
| IDW | Inverse distance weighting |
| IM-PCM | Interpolation method based on the parameter contour map |
| KMA | Kora Meteorological Administration |
| LOGN | Log-normal distribution |
| NOAA | National Oceanic and Atmospheric Administration |
| NSE | Nash-Sutcliffe efficiency |
| POT | Peak over threshold |
| $r$ | Pearson correlation coefficient |
| RMSE | Root mean square error |
| QM | Quantile mapping |
| WEI | Weibull distribution |
| 20CR | The 20th century reanalysis by the NOAA |

## Appendix B. List of Symbols

| ID | Definitions |
|---|---|
| RAW | Uncorrected ERA-20c daily precipitation |
| TH | Cut-off threshold for quantile mapping (QM) approach |
| $\alpha$ | shape parameter of a gamma distribution |





| $\beta$ | scale parameter of a gamma distribution |
|---|---|
| $\xi$ | Shape parameter of a GPD |
| $\theta$ | Scale parameter of a GPD |
| $u$ | High upper threshold for a GPD |



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





# Tables

*Table 1. The local rainfall stations used in this study*

| Station No. | Name | Latitude (°N) | Longitude (°E) | Elevation(m. asl) | Annual rainfall(mm)* |
|---|---|---|---|---|---|
| St. 1 | Sokcho | 38.2508 | 128.5644 | 19.5 | 1,374.6 |
| St. 2 | Daegwallyeong | 37.6769 | 128.7181 | 774.0 | 1,736.4 |
| St. 3 | Chuncheon | 37.9025 | 127.7356 | 79.1 | 1,304.9 |
| St. 4 | Gangneung | 37.7514 | 128.8908 | 27.4 | 1,436.6 |
| St. 5 | Seoul | 37.5714 | 126.9656 | 11.1 | 1,386.8 |
| St. 6 | Incheon | 37.4775 | 126.6247 | 69.6 | 1,183.0 |
| St. 7 | Wonju | 37.3375 | 127.9464 | 150.0 | 1,318.6 |
| St. 8 | Suwon | 37.2700 | 126.9875 | 38.3 | 1,274.9 |
| St. 9 | Chungju | 36.9700 | 127.9525 | 116.5 | 1,202.0 |
| St. 10 | Seosan | 36.7736 | 126.4958 | 30.3 | 1,254.9 |
| St. 11 | Cheongju | 36.6361 | 127.4428 | 58.6 | 1,229.7 |
| St. 12 | Daejeon | 36.3689 | 127.3742 | 70.3 | 1,353.0 |
| St. 13 | Chupungyeong | 36.2197 | 127.9944 | 246.1 | 1,171.5 |
| St. 14 | Andong | 36.5728 | 128.7072 | 141.5 | 1,017.3 |
| St. 15 | Pohang | 36.0325 | 129.3794 | 3.7 | 1,145.4 |
| St. 16 | Gunsan | 36.0019 | 126.7631 | 24.6 | 1,210.8 |
| St. 17 | Daegu | 35.8850 | 128.6189 | 65.5 | 1,047.0 |
| St. 18 | Jeonju | 35.8214 | 127.1547 | 54.8 | 1,291.6 |
| St. 19 | Ulsan | 35.5600 | 129.3200 | 36.0 | 1,265.5 |
| St. 20 | Gwangju | 35.1728 | 126.8914 | 73.8 | 1,387.9 |
| St. 21 | Busan | 35.1044 | 129.0319 | 71.0 | 1,500.2 |
| St. 22 | Mokpo | 34.8167 | 126.3811 | 39.4 | 1,139.4 |
| St. 23 | Yeosu | 34.7392 | 127.7406 | 66.0 | 1,420.1 |
| St. 24 | Jinju | 35.1636 | 128.0400 | 31.6 | 1,504.8 |
| St. 25 | Yangpyeong | 37.4886 | 127.4944 | 49.4 | 1,359.6 |
| St. 26 | Icheon | 37.2639 | 127.4842 | 79.4 | 1,330.9 |
| St. 27 | Inje | 38.0600 | 128.1669 | 201.6 | 1,167.8 |
| St. 28 | Hongcheon | 37.6833 | 127.8803 | 142.3 | 1,353.2 |
| St. 29 | Jecheon | 37.1592 | 128.1942 | 265.0 | 1,345.8 |
| St. 30 | Boeun | 36.4875 | 127.7339 | 176.4 | 1,275.0 |
| St. 31 | Cheonan | 36.7794 | 127.1211 | 24.0 | 1,229.4 |
| St. 32 | Boryeong | 36.3269 | 126.5572 | 16.9 | 1,219.6 |
| St. 33 | Buyeo | 36.2722 | 126.9206 | 12.7 | 1,323.3 |
| St. 34 | Geumsan | 36.1056 | 127.4817 | 171.7 | 1,277.1 |
| St. 35 | Buan | 35.7294 | 126.7164 | 13.4 | 1,249.8 |
| St. 36 | Imsil | 35.6122 | 127.2853 | 249.3 | 1,340.2 |
| St. 37 | Jeongeup | 35.5631 | 126.8658 | 46.0 | 1,317.1 |
| St. 38 | Namwon | 35.4053 | 127.3328 | 91.7 | 1,351.0 |
| St. 39 | Jangheung | 34.6886 | 126.9194 | 46.4 | 1,493.7 |
| St. 40 | Haenam | 34.5533 | 126.5689 | 14.4 | 1,322.4 |
| St. 41 | Goheung | 34.6181 | 127.2756 | 54.5 | 1,459.2 |





| St. 42 | Yeongju | 36.8717 | 128.5167 | 212.2 | 1,268.1 |
|--------|---------|---------|----------|-------|---------|
| St. 43 | Mungyeong | 36.6272 | 128.1486 | 172.0 | 1,241.5 |
| St. 44 | Uiseong | 36.3558 | 128.6883 | 83.2 | 1,016.5 |
| St. 45 | Gumi | 36.1306 | 128.3206 | 50.3 | 1,051.1 |
| St. 46 | Yeongcheon | 35.9772 | 128.9514 | 95.0 | 1,039.3 |
| St. 47 | Geochang | 35.6711 | 127.9108 | 222.4 | 1,298.9 |
| St. 48 | Sancheong | 35.4128 | 127.8789 | 0.8 | 1,512.7 |

* Annual mean precipitation estimated from 1973 to 2010





*Table 2. The selected distributions among six distributions based on AIC and BIC values for the extremes from observed and ERA-20c daily precipitation over the 95$^{th}$ and 99$^{th}$ percentiles for all 48 stations*

| Percentile | Data | GPD | GEV | LOGN | WBL | GUM | GAM |
|---|---|---|---|---|---|---|---|
| 95th | Observation | 47 | 1 | 0 | 0 | 0 | 0 |
| | ERA-20c | 48 | 0 | 0 | 0 | 0 | 0 |
| 99th | Observation | 47 | 1 | 0 | 0 | 0 | 0 |
| | ERA-20c | 47 | 1 | 0 | 0 | 0 | 0 |





*Table 3. Comparisons of root-mean-square-error (RMSE) and Nash-Sutcliffe efficiency (NSE) between the observed and the corrected ERA-20c for different thresholds [TH1 (>0mm/day), TH2 (>0.1mm/day), TH3 (>1mm/day) and TH4 (Frequency adjustment)] and the uncorrected ERA-20c precipitation.*

| Data | Measures | TH1 | TH2 | TH3 | TH4 | ERA-20c |
|---|---|---|---|---|---|---|
| **Monthly mean (mm/month)** | RMSE (mm) | 119.24 | 110.50 | 42.57 | 4.77 | 15.59 |
| | NSE | -0.899 | -0.631 | 0.758 | 0.997 | 0.968 |
| **10-days running mean. (mm/day)** | RMSE (mm) | 4.03 | 3.74 | 1.49 | 0.51 | 0.56 |
| | NSE | -0.886 | -0.622 | 0.744 | 0.970 | 0.963 |





*Table 4. A comparison of the mean values between the observed and modelled data (i.e. the corrected ERA-20c by gQM, gpQM95 and gpQM99, and the uncorrected ERA-20c)*

| Data | Measures | gQM | gpQM95 | gpQM99 | ERA-20c |
|---|---|---|---|---|---|
| **Monthly mean (mm/month)** | RMSE (mm) | 4.77 | 9.41 | 5.12 | 15.59 |
| | NSE | 0.997 | 0.988 | 0.997 | 0.968 |
| **10-days running mean. (mm/day)** | RMSE (mm) | 0.507 | 0.545 | 0.497 | 0.563 |
| | NSE | 0.970 | 0.966 | 0.971 | 0.963 |



*Table 5. A comparison of the mean values between the observed and the modelled precipitation for three different approaches by using an set of parameters interpolated from IM-PCM within the leave-one-out cross validation framework*

| Data | Measures | gQM | gpQM95 | gpQM99 | ERA-20c |
|---|---|---|---|---|---|
| **Monthly mean (mm/month)** | RMSE (mm) | 4.14 | 10.31 | 5.27 | 15.59 |
| | NSE | 0.998 | 0.986 | 0.996 | 0.968 |
| **10-days running mean. (mm/day)** | RMSE (mm) | 0.502 | 0.562 | 0.498 | 0.563 |
| | NSE | 0.971 | 0.963 | 0.971 | 0.963 |



Table 6. Pearson correlation coefficients(r) between elevations and parameters for gQM, gpQM95 and gpQM99 for all 48 stations

| Bias Correction Methods | Parameter | Gamma Distribution | | | | | | | | | | | | GPD | |
|---|---|---|---|---|---|---|---|---|---|---|---|---|---|---|---|
| | | r | | | | | | | | | | | | Parameter | r |
| | | Jan | Feb | Mar | Apr | May | Jun | Jul | Aug | Sep | Oct | Nov | Dec | | |
| gQM | α | -0.40 | -0.14 | 0.06 | 0.18 | 0.07 | 0.16 | 0.22 | 0.06 | 0.15 | 0.00 | -0.06 | -0.14 | ξ | - |
| gpQM95 | | -0.37 | -0.13 | 0.05 | 0.17 | 0.09 | 0.19 | 0.26 | 0.09 | 0.15 | 0.12 | -0.13 | -0.18 | | -0.01 |
| gpQM99 | | -0.40 | -0.14 | 0.06 | 0.18 | 0.07 | 0.16 | 0.24 | 0.08 | 0.14 | 0.03 | -0.08 | -0.14 | | -0.05 |
| gQM | β | 0.09 | -0.15 | -0.25 | -0.22 | -0.14 | -0.20 | -0.11 | -0.11 | -0.02 | 0.17 | -0.02 | -0.11 | θ | - |
| gpQM95 | | 0.02 | -0.16 | -0.22 | -0.23 | -0.20 | -0.25 | -0.18 | -0.14 | -0.08 | -0.10 | 0.02 | -0.08 | | -0.05 |
| gpQM99 | | 0.09 | -0.14 | -0.25 | -0.23 | -0.17 | -0.21 | -0.13 | -0.16 | -0.03 | 0.09 | -0.03 | -0.11 | | -0.01 |



## *Figures*

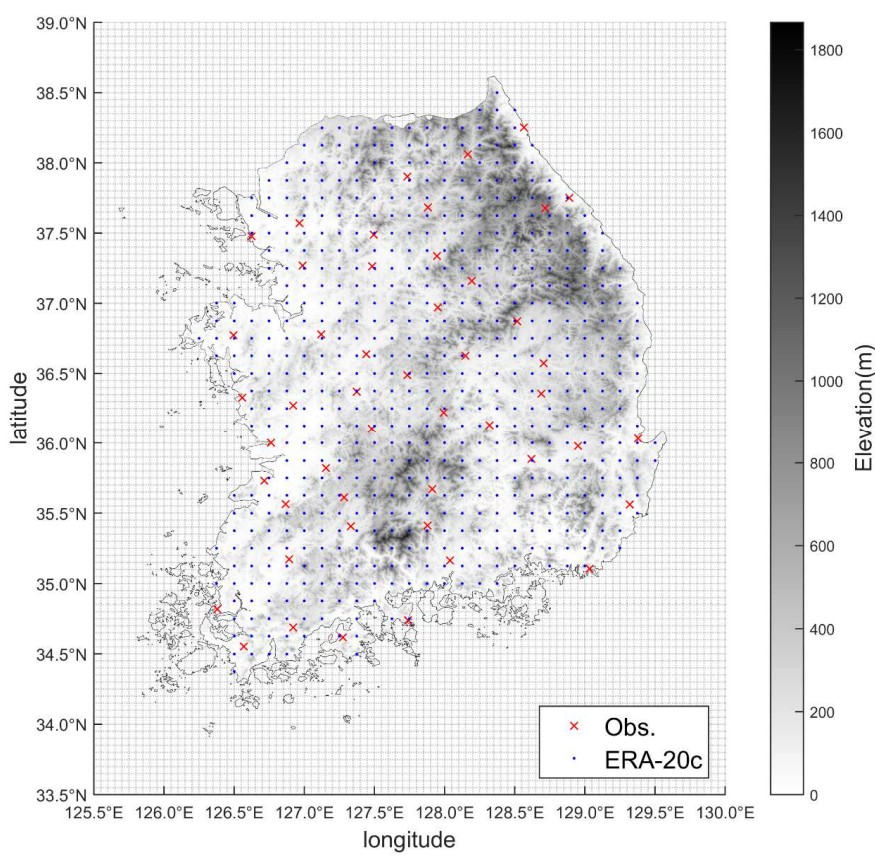

Figure 1. A map showing the study area, local gauging stations and grid points of ERA-20c. The grey shading on the map indicates elevations





*(a)*

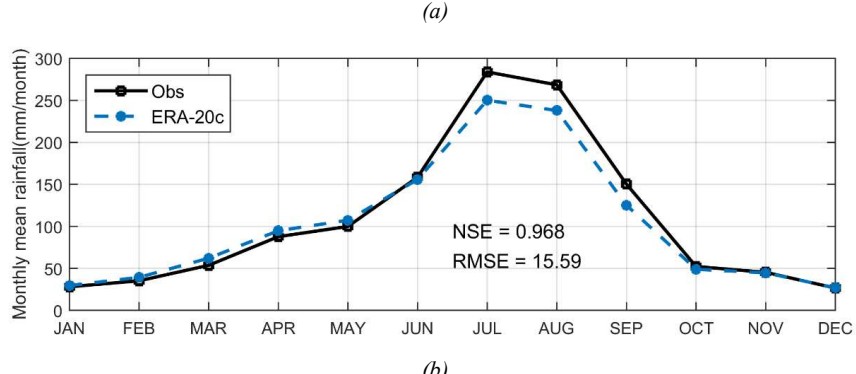

*(b)*

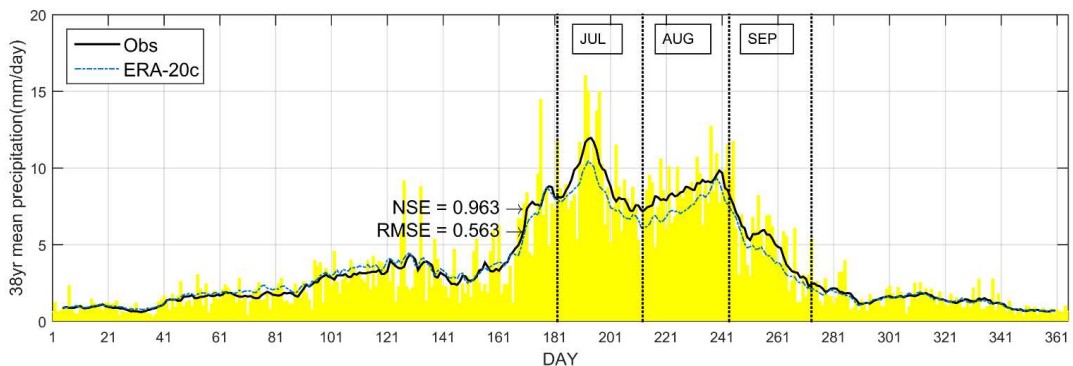

5    *Figure 2. A comparison of the mean values of ERA-20c daily precipitation on the annual basis. (a) Monthly mean*
*comparison between the observed (Obs) and ERA-20c, and (b) observed 38-year (1973-2010) mean of daily*
*precipitation (yellow bar) and its 10-day running mean (black solid line) along with 10-day running mean*
*estimated from ERA-20c (blue dotted line) for all 48 stations*




*(a)*

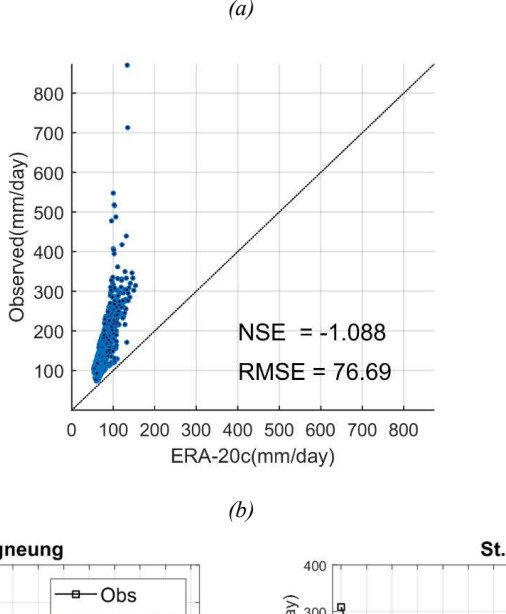

*(b)*

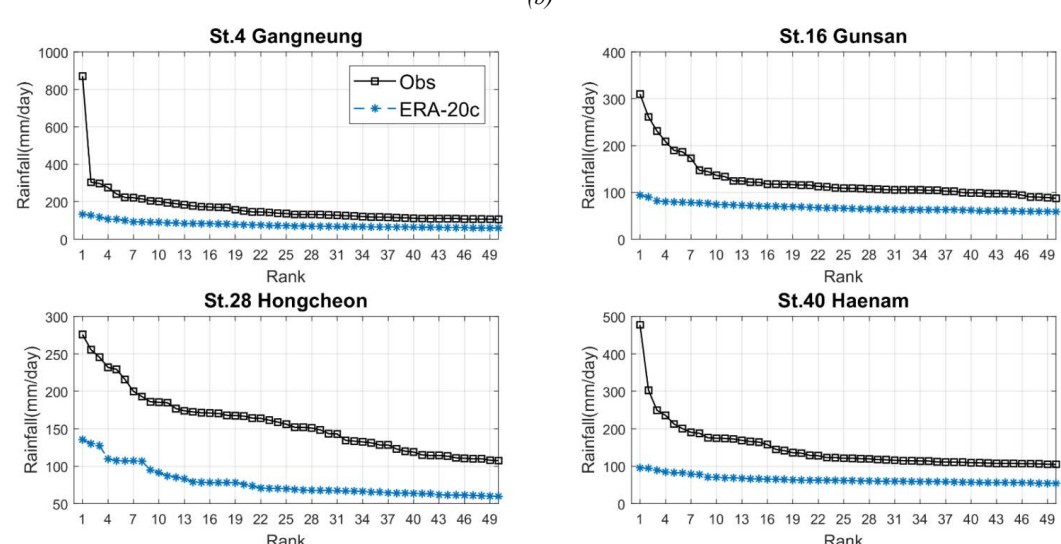

5   *Figure 3. Evaluation of bias associated with 50 top extreme rainfall events. (a) Scatter plot of the extremes between the observed and ERA-20c over the entire region of interest and (b) comparison of the deviation corresponding to the rank for the station 4, 16, 28 and 40 for the baseline period 1973-2010.*




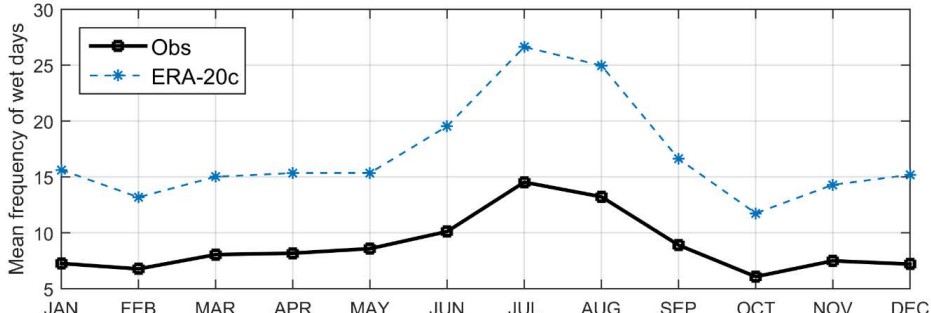

*Figure 4.Monthly wet-day frequency for the observed (black solid line) and ERA-20c (blue dotted line) for all 48 stations for the baseline period (1973-2010).*

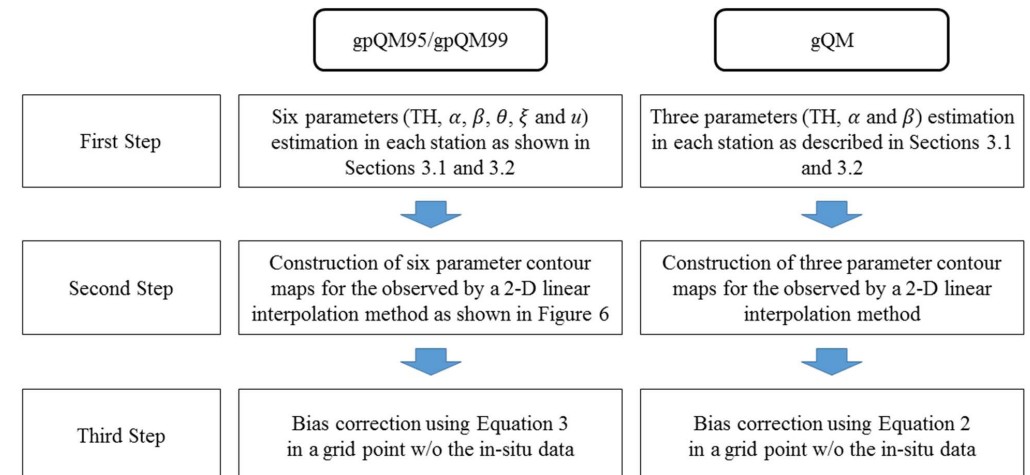

*Figure 5. A flowchart of the proposed quantile mapping approaches (gpQM95/gpQM99 and gQM) based on the parameter contour maps (IM-PCM)*





*(a)*

*(b)*

*(c)*          *(d)*

*Figure 6. Parameter contour maps for gpQM99 approach. (a) Maps of shape (α) and scale (β) parameter of the gamma distribution in August, (b) maps of shape (ξ) and scale (θ) parameter of the GPD, (c) map of frequency of wet-days corresponding to the cut-off threshold (TH) in August, and (d) maps of upper threshold (u) for the GPD. Here, the GPD is applied to entire POTs on an annual basis.*



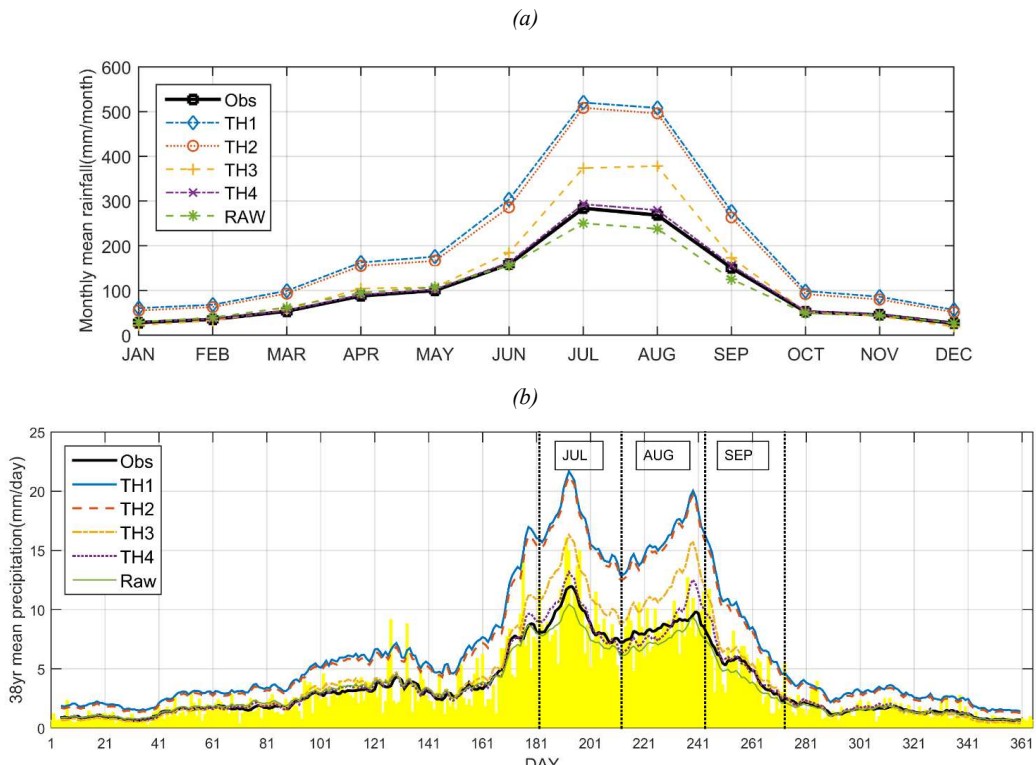

5    *Figure 7. A comparison of mean rainfall between the observation and the corrected ERA-20c with different*
*thresholds [TH1(>0mm/day), TH2(>0.1mm/day), TH3(>1mm/day) and TH4(Frequency adjustment)] and the*
*uncorrected ERA-20c (RAW)) on the annual basis. All values are averaged over all 48 stations from 1973 to 2010.*
*(a) Monthly mean comparison between different thresholds and (b) observed 38-year (1973-2010) mean of daily*
*precipitation (yellow bar) and its 10-day running mean (black solid line), along with a set of 10-day running means*
10    *estimated from bias corrected ERA-20c daily precipitations using four different thresholds for all 48 stations.*





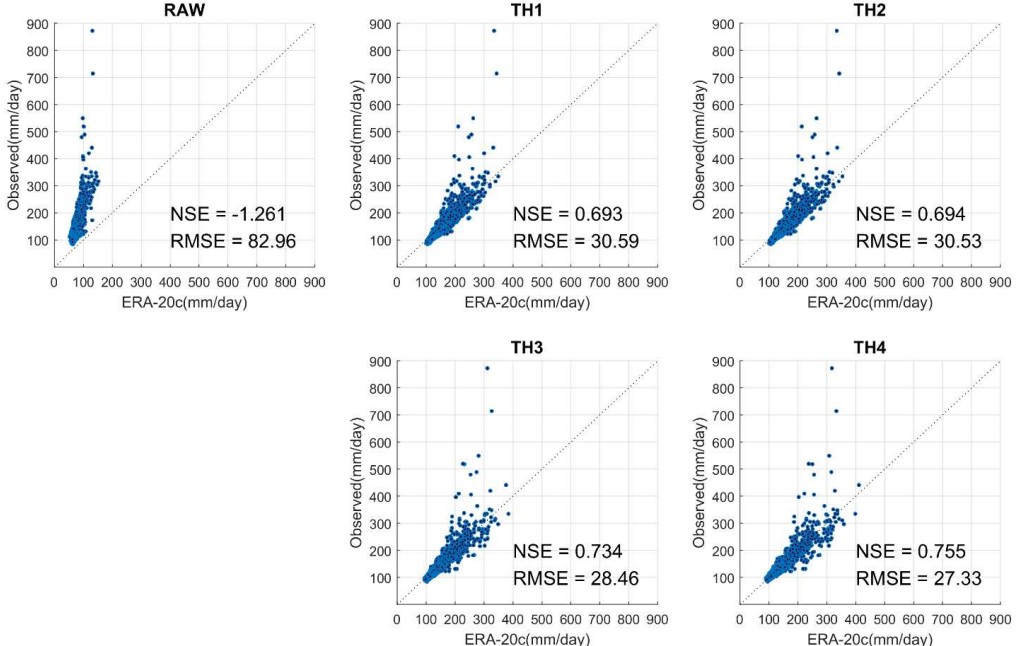

*Figure 8. Scatter plots between the observed and the modelled extreme rainfalls associated with different thresholds over the 99<sup>th</sup> percentile for all 48 stations. RAW indicates the uncorrected ERA-20c and the others represent the results from the corrected ERA-20c by gQM with different thresholds [TH1(>0mm/day),TH2(>0.1mm/day), TH3(>1mm/day) and TH4(Frequency adjustment)].*





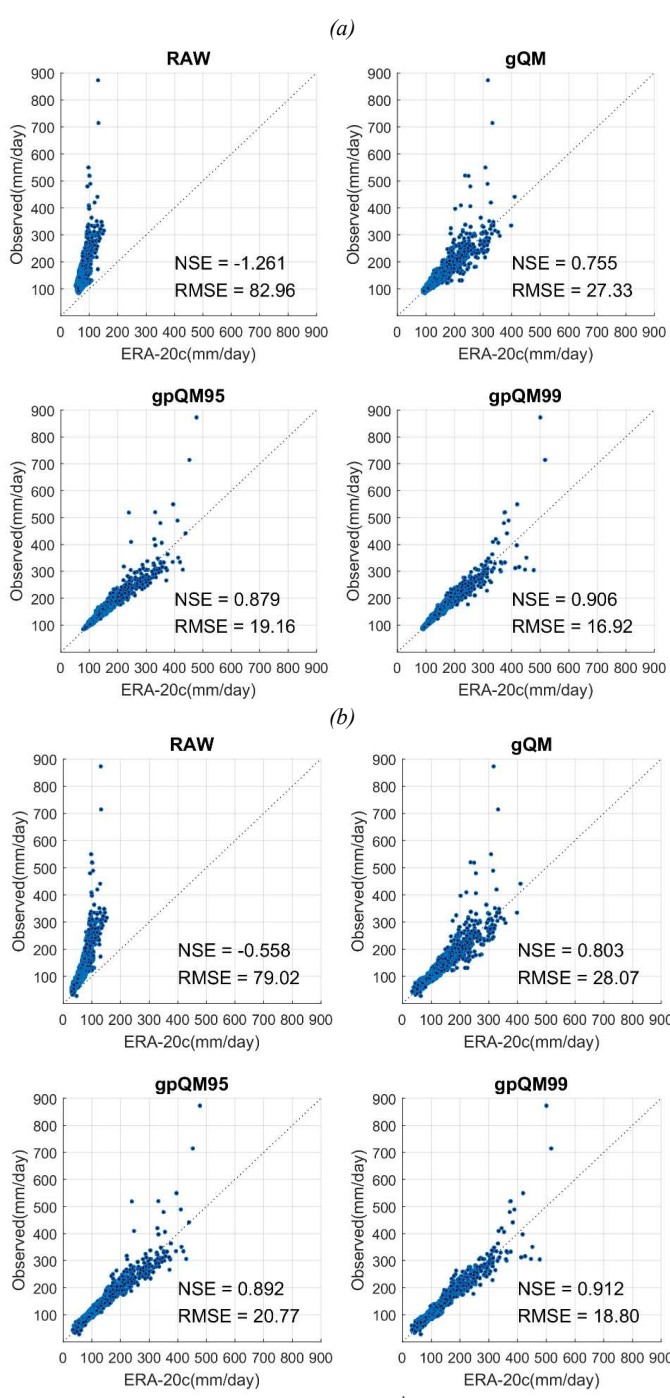

5    *Figure 9. Scatter plots for (a) the extreme rainfalls over the 99$^{th}$ percentile and (b) annual maximum series (AMS)*

*extracted from the observed and the bias corrected ERA-20c daily precipitation over all 48 stations*



*(a)*

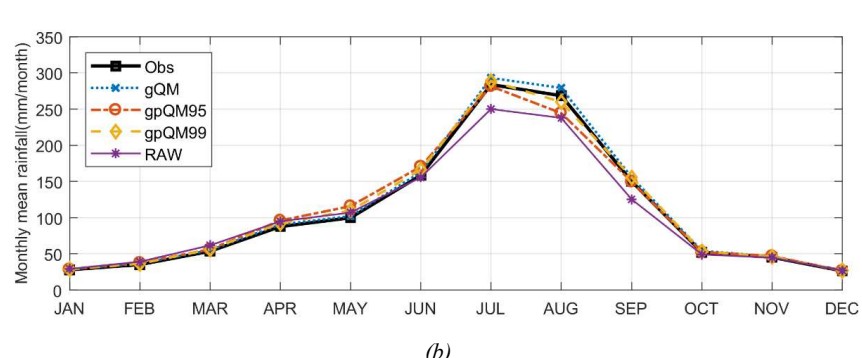

*(b)*

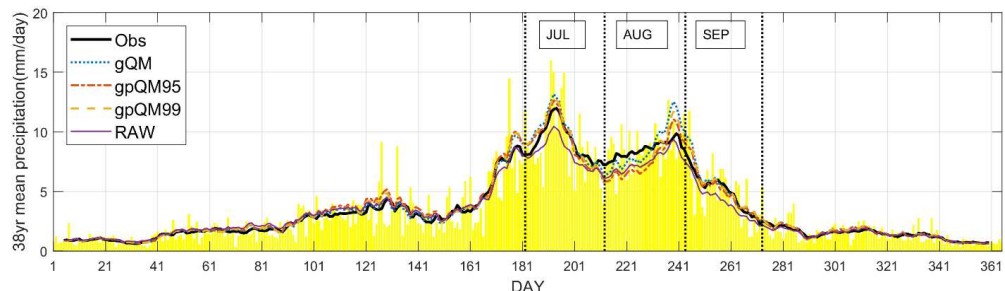

*Figure 10. A comparison of mean rainfall between the observation and the corrected ERA-20c with different QM approaches. (a) Monthly mean comparison between different QMs and (b) observed 38-year (1973-2010) mean of daily precipitation (yellow bar) and its 10-day running mean (black solid line), along with a set of 10-day running means estimated from bias corrected ERA-20c daily precipitations using three different QM approaches for all 48 stations.*





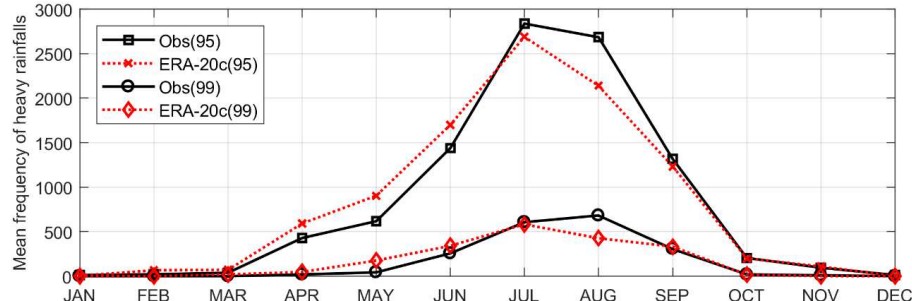

*Figure 11. Monthly mean frequency of the heavy rainfalls over the 95th and 99th percentile from the observed (Obs) and ERA-20c daily precipitation. Here, the mean frequency is averaged over 48 stations from 1973 to 2010*





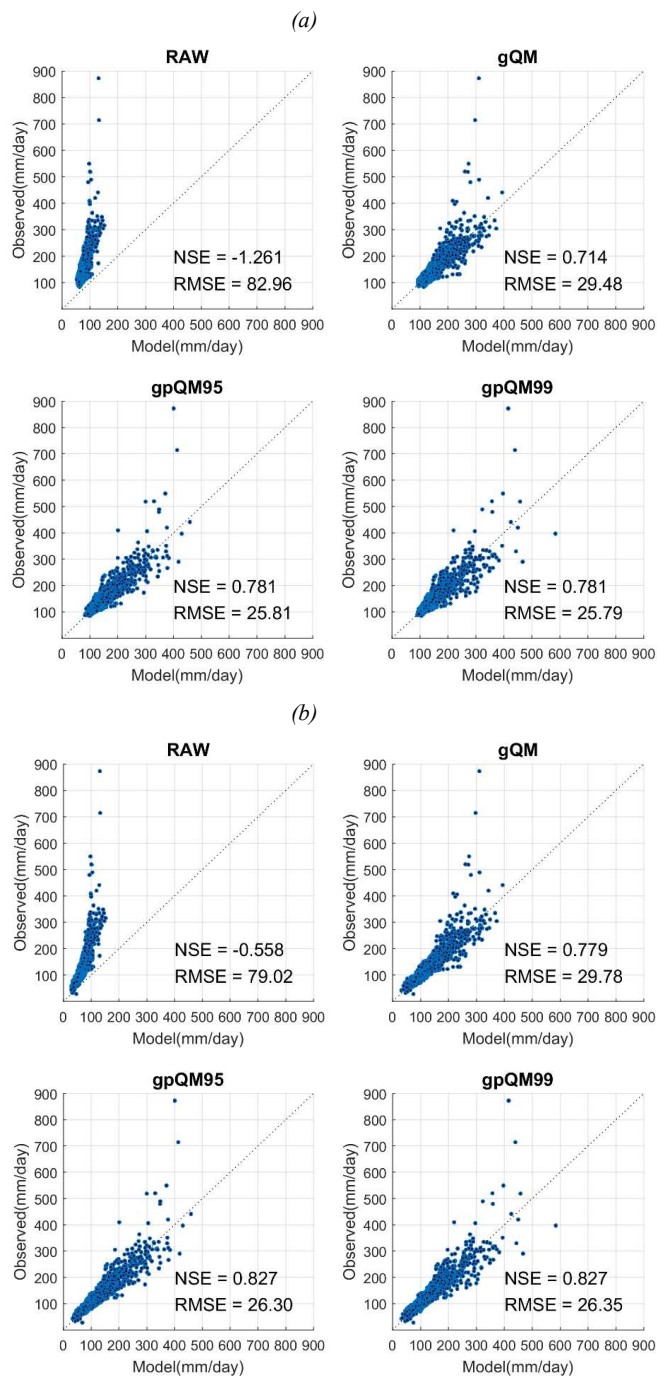

Figure 12. Scatter plots for (a) the extreme rainfalls over the 99$^{th}$ percentile and (b) annual maximum series (AMS) extracted from the observed and the bias corrected ERA-20c daily precipitation over all 48 stations. All the results presented here are obtained by leave-one-out cross validation.





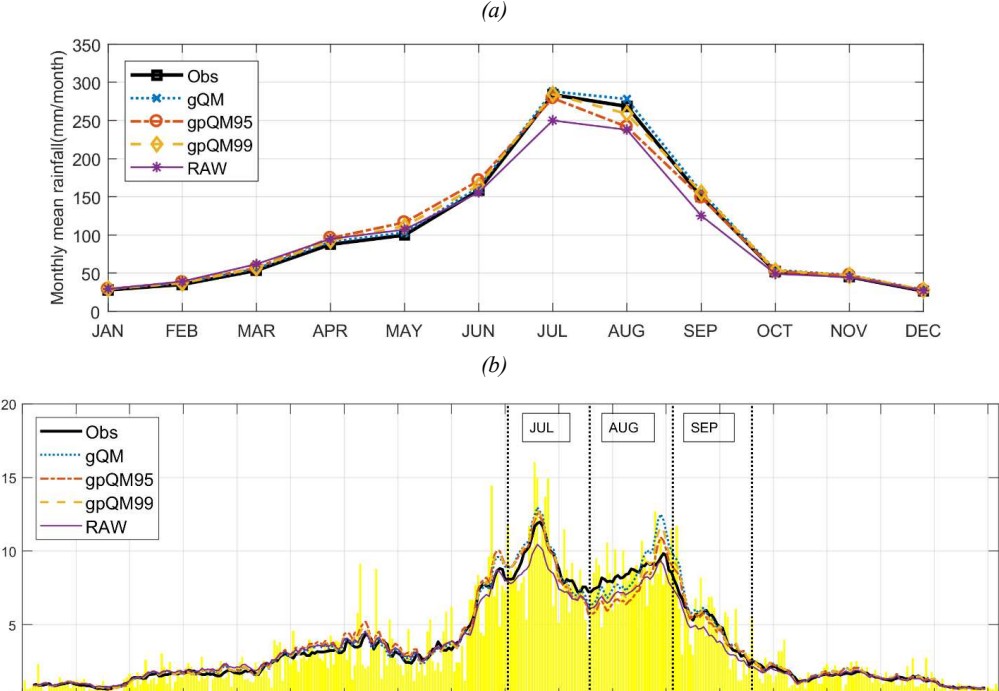

5    *Figure 13. A comparison of cross validation results for the mean rainfall between the observation and the corrected*
*ERA-20c with different QM approaches. (a) Monthly mean comparison between different QMs and (b) observed 38-*
*year (1973-2010) mean of daily precipitation (yellow bar) and its 10-day running mean (black solid line), along with*
*a set of 10-day running means estimated from bias corrected ERA-20c daily precipitations using three different QM*
*approaches for all 48 stations. All the results presented here are obtained by leave-one-out cross validation.*





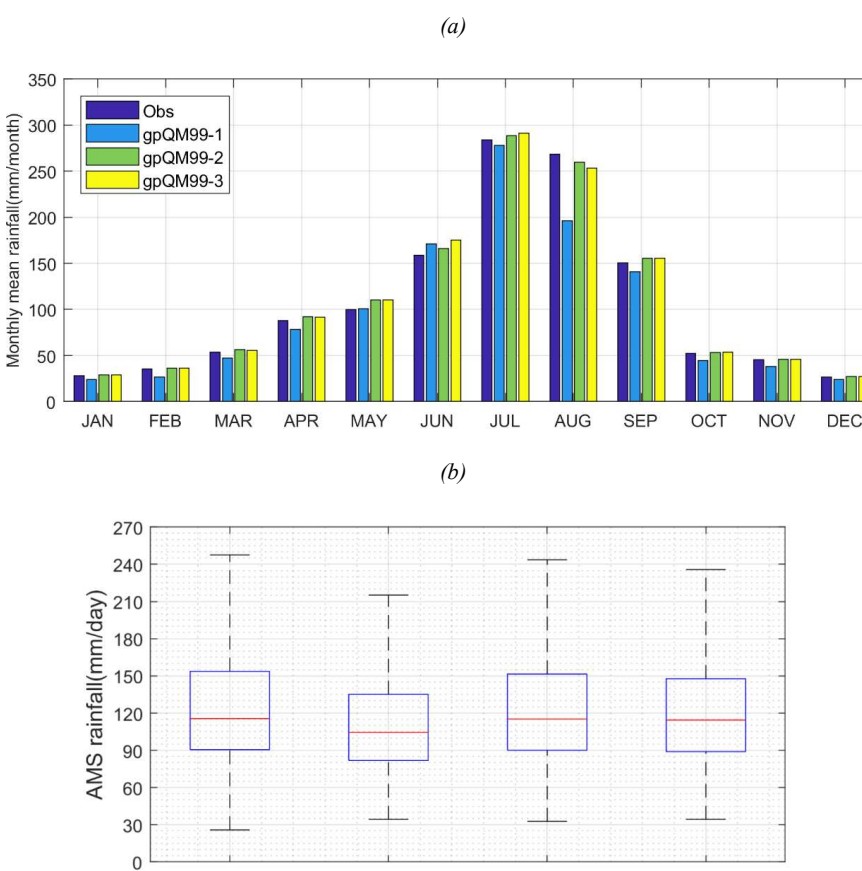

5    *Figure 14. A retrospective analysis for a comparison between the observed precipitation (1973-2010) and the*
     *corrected ERA-20c by gpQM99 with three different periods:1900-1972 (gpQM99-1), 1973-2010 (gpQM99-2) and*
     *1900-2010 (gpQM99-3). (a) Monthly mean rainfalls and (b) box plot of the annual maximum series (AMS)*
     *rainfalls.*