# Peer review of "Exploring the Long-Term Reanalysis of Precipitation and the Contribution of Bias Correction to the Reduction of Uncertainty over South Korea: A Composite Gamma-Pareto Distribution Approach to the Bias Correction"

_Hydrology and Earth System Sciences, 2018_

## Referee Comment (RC1) · Anonymous Referee #1 · 8 Mar 2018

This is an interesting article investigating and bias-correcting the long-term reanalysis of precipitation over South Korea. Different combination of transfer function and wet frequency adjustment methods are applied to correct the precipitation time series. Explicit analysis was down and detailed results were shown. The manuscript is well presented, however, there are still spaces to improve.

Genreal comments: 1. QM based methods are fundamental tool for bias correction of

climate variables. However, Many similar studies have been done. It is also quite normal to take the combination fo different transfer functions to describe CDF for different quantiles. Thus, the contribution of this paper to the scientific progress is low. Also, the title of this manuscript highlights the"Contribution of Bias Correction to the Reduction of Uncertainty", but it is not well explored in this paper.

2. Unlike the temperature, bias correction of precipitation is more challenging due to the fact of spatial/temporal heterogeneity and zero inflation. The bias correction should take care of all the four cases: (0,0), (0,1), (1,0) and (1,1), where 0 denotes a dry day and 1 indicates a wet day. It is not clear how the authors did for there four different cases.

3. The authors proposed a new framework for bias correction in un-gauged area by using the IM-PCM method to interpolate the parameter of transfer functions. To my opinions, the contour mapping technique could bring large uncertainties and biases. Even the the results are validated, but it is based on the average of all the 48 stations. The study of spatial impacts of this technique on the bias correction is still missing.

4. It is not clear that how the authors set the calibration and validation period. It seems the complete study period is used for both calibration and validation.

Specific comments: Page 4, Line 5: "but not bias correction issues" -> "but not bias correction technique issues" Page 4, Line 11: In addition to linear scaling, local intensity, power transformation, and quantile mapping, there are also there sophisticated method for bias correction, e.g. copula-based technique. Page 5, Line 9: "Comparatively little attention has been given to the bias correction of the reanalysis data" is not correct. There is no clear clue for this. Page 10, Line 20: "(TH4). The frequency" -> "(TH4), the frequency" Page 16, Line 9: It is not clear if all the TH are tested together with gQM or not? Page 17, Line 13: "TH4 performs the best with 0.755 for NSE". It is a bit confuse to me, as the TH only affects the wet frequency of the time series. How it affects the extreme correction? This needs to be explained in more detail. Page 17,

Line 20: It is not clear if the TH4 is take for gpQM tests.

---

## Referee Comment (RC2) · Anonymous Referee #2 · 6 Apr 2018

Review of "Exploring the Long-Term Reanalysis of Precipitation and the Contribution of Bias Correction to the Reduction of Uncertainty over South Korea: A Composite Gamma-Pareto Distribution Approach to the Bias Correction" by Kim et al.

The authors present and evaluate a bias correction of the ECMWF ERA-20c reanalysis for South Korea. The correction is based on a parametric quantile mapping and calibrated between reanalysis grid-box and observed station precipitation, and extended

to the full field by interpolating the transfer function parameters in space.

I cannot recommend this manuscript for publication. At least major parts should be substantially revised, and the spatial model should be fully omitted. My major concerns are as follows:

1. Deterministic bias correction of precipitation cannot be used for downscaling, and in particular not to create spatial fields. Maraun (2013) has demonstrated that bias correction suffers from the same conceptual flaw as inflated regression. Differences between reanalyis and station observations (in particular the magnitudes of summer extremes and wet day frequencies) are not necessarily biases, but to a substantial degree due to the scale gap between the area average of the reanalysis and the point-scale of the observations. Local-scale variability is not fully determined by the grid box average, a deterministic rescaling as done by quantile mapping cannot create the missing local variability. Instead the large-scale variability is inflated. Thus, the corrected time series have similar marginal properties as the local observations, but do not have the correct spatial-temporal properties. This is a problem in particular for spatial fields, as the spatial distribution of the corrected field is still that of the reanalysis (apart from the wet-day correction), but only inflated. It does not represent the small-scale variability of summer thunderstorms, e.g. The problem is severe for extreme events: dry areas as well as the magnitude of precipitation falling over a certain area are substantially overestimated (Maraun 2013). Thus, using these data for hydrological modeling would likely result in dangerously misleading results. This issue is rather irritating, given that the authors cite Volosciuk et al. (2017) who discuss this issue in depth. In fact, the only correct solution would be a stochastic bias correction that bias corrects (if needed at all in this case) and additionally adds random small-scale variability (either in a single-site approach as suggested by Volosciuk et al. (2013) or with a fully spatial model. If only single locations are considered (without using time series at multiple sites), a quantile mapping to the point-scale might be justified based on pragmatic reasoning.
A major problem of the manuscript is that the evaluation is essentially blind to these problems. They are mostly visible in the spatial and interannual variability. None of these aspects have been evaluated.

2. The discussion in the manuscript is rather naive and largely ignores problems of bias correction and reanalysis data. It also ignores much of the literature in the field. For instance, it is well known that at least the first versions of century-long reanalysis data strongly misrepresent long-term climatic trends, or that synoptic-scale variability in the Tropics is only weakly constraint in reanalysis data (Krueger et al, 2013; Befort et al., 2016; Brands et al., 2012). These issues are not discussed in the manuscript. Similarly, the downscaling issues discussed above have not been acknowledged, differences between biases and scale-gaps in the given example have not been discussed. In fact, the authors do not make any attempt to discuss which kind of biases can be corrected in their context. E.g., misrepresented long-term trends, spatial-temporal variability (apart from wet-day corrections) or a misrepresented tropical day-to-day variability will not be corrected by the bias correction. See, e.g., Maraun et al., 2017, for a discussion of several issues (many are relevant in a climate change context, but some apply also here).

3. The language needs substantial revision, as well as the logic within several sentences. I will give some examples below.

Further comments:

p2 l11: this sentence makes no sense and does not logically link to the previous sentence.

p2 l16: the data are not just coarsely represented in model calibration, they are simply coarse.

p2 l23: what does "finer" refer to? Or should it be just "fine"? In any case I would not agree that reanalysis are provided at a fine resolution. What is more important is that

they provide a complete field.

p3 l2: "spans from" English!

p3 l11-13: this does not make sense. If pressure and wind are not assimilated, how can the synoptic situation then be represented?

p3 l14: what does "on the other hand" refer to?

p3: here the limitations of the reanalysis data should be discussed.

p4 l9: there is a more recent review by Maraun (2016) and the recent book by Maraun and Widmann (2018). Also the selection of methods is rather arbitrary.

p4 l13: bias correction cannot reduce errors in numerical models! It can, at best, postprocess numerical models.

p4 l14: "Jacob Themessl et al" should be "Themessl et al.". The name is Matthias J. Themessl.

p4 l15 "referred to as other names" grammar!

p4 l18 "usually based on a gamma". No - this is not true. There are many other implementations, and often non-parametric approaches are used.

p5 l1: "underestimation" Not necessarily. In particular moderate extremes might be overestimated (in the range where the scale parameter dominates).

p5 l13 and following: as discussed above, this approach is not sensible, at least not for a deterministic method which is interpreted at multiple sites.

p12 l12-16: this listing is a bit naive. The GEV is designed to model block maxima. It may fit a distribution tail rather well because it is flexible (3 parameters), but conceptually this doesn't make sense. Here some discussion should be added.

p13, eq. (3): this model is a bit crude. There are many implementations that ensure at least continuity at the transition point between gamma and GPD, some even smoothness. The method here essentially has a jump.

p14, l2: "mainly" well, what other reason should there be?

Section 3.3: as discussed, this is extremely dangerous and should not be done.

References:

Befort, D. J., Wild, S., Kruschke, T., Ulbrich, U. and Leckebusch, G. C. (2016), 'Different long- term trends of extra-tropical cyclones and windstorms in ERA-20C and NOAA-20CR reanalyses', Atmos. Sci. Lett. 17(11), 586–595.

Brands, S., Gutiérrez, J. M. and Herrera, S. (2012), 'On the use of reanalysis data for downscaling', J. Climate 25, 2517–2526.

Krueger, O., Schenk, F., Feser, F. and Weisse, R. (2013), 'Inconsistencies between long- term trends in storminess derived from the 20CR reanalysis and observations', J. Climate 26(3), 868–874.

Maraun, D. (2013), 'Bias correction, quantile mapping and downscaling: revisiting the inflation issue', J. Climate 26, 2137–2143.

Maraun, D. (2016), 'Bias correcting climate change simulations - a critical review', Curr. Clim. Change Rep. 2(4), 211–220.

Maraun, D., Shepherd, T. G., Widmann, M., Zappa, G., Walton, D., Gutierrez, J. M., Hage- mann, S., Richter, I., Soares, P. M. M., Hall, A. and Mearns, L. (2017b), 'Towards process-informed bias correction of climate change simulations', Nat. Clim. Change, online first, DOI 10.1038/nclimate3418.

Maraun, D., and Widmann, M. (2018), 'Statistical Downscaling and Bias correction for Climate Research ', Cambridge University Press.

Volosciuk, C., Maraun, D., Vrac, M. and Widmann, M. (2017), 'A combined statistical bias correction and stochastic downscaling method for precipitation', Hydrol. Earth

Syst. Sci. 21(3), 1693–1719.

---

## Author Comment (AC1) · 11 May 2018

**Authors' response to Referee #1**

For clarity, authors' responses are presented by blue colour.

We have answered all the comments of the reviewer 1. Answers are attached to this revision note. Along with the answers we are also explaining all the changes we have done.

This is an interesting article investigating and bias-correcting the long-term reanalysis of precipitation over South Korea. Different combination of transfer function and wet frequency adjustment methods are applied to correct the precipitation time series. Explicit analysis was done and detailed results were shown. The manuscript is well presented, however, there are still spaces to improve.

**(Response)** Thank you for the comments.

General comments:

1. QM based methods are fundamental tool for bias correction of climate variables. However, Many similar studies have been done. It is also quite normal to take the combination of different transfer functions to describe CDF for different quantiles. Thus, the contribution of this paper to the scientific progress is low.

**(Response)** The composite distribution based on the quantile mapping approach has been recently employed to describe the distinctive feature of CDF over different quantiles, especially for the correction of climate change scenarios (Gutjahr and Heinemann, 2013; Nyunt et al., 2016; Smith et al., 2014; Volosciuk et al., 2017). However, the bias correction of the reanalysis data using the composite distribution has not been explored sufficiently. It appears that only a very limited number of studies have investigated the validity of the different transfer function method for the multi-decadal reanalysis data (Teng et al., 2015; Vrac and Naveau, 2007), but not for century-long reanalysis such as ERA-20c. From this perspective, the composite distribution addresses the context of the bias correction to facilitate the improved transfer function, particularly new for the reanalysis (or numerical weather prediction data).

Moreover, the previous studies have applied QM approaches to gauged catchments, where a statistical relationship between the observed and modelled can be established. However, the suggested QM algorithm in this study can significantly reduce the systematic error associated with daily precipitation, which has not been improved in regional-scale studies, over the spatio-temporally limited rainfall observation network. More specifically, this study first explored the

ERA-20c data and corrected them in 603 grid points using dozens of gauging stations over South Korea for the reference period 1973-2010. This QM approach can also be applied to the whole of the 20$^{th}$ century (1900-2010) without synthesizing the corresponding observation, albeit the study has been carried out in South Korea for the reference period. Thus, we believe that the methodology used in this study is novel and contributes positively to the hydrological community.

2. Also, the title of this manuscript highlights the "Contribution of Bias Correction to the Reduction of Uncertainty", but it is not well explored in this paper.

 (Response) As indicated, this study mainly focused on the bias correction of ERA-20c daily precipitation, and we considered that the century-long dataset could contribute to the reduction of the uncertainty in hydrological analysis where a limited number of observations were generally given. In this study, we only explored the uncertainty range for three different periods. Thus, we changed the title of this manuscript to "Exploring the Long-Term Reanalysis of Precipitation and its Bias Correction using a Composite Gamma-Pareto Distribution Approach over South Korea" upon your comments. We will further explore the uncertainty reduction in using the long-term reanalysis data in the next study.

3. Unlike the temperature, bias correction of precipitation is more challenging due to the fact of spatial/temporal heterogeneity and zero inflation. The bias correction should take care of all the four cases: (0,0), (0,1), (1,0) and (1,1), where 0 denotes a dry day and 1 indicates a wet day. It is not clear how the authors did for there four different cases.

(**Response**) Thanks for good comments. It is expected that the temporal correlation of ERA-20c daily precipitation would be high enough to compare the rainfall sequences. However, it has been acknowledged that the temporal correlation is relatively low to directly compare the rainfall sequences (Poli et al., 2013). In our study, for instance, the range of Pearson's correlation coefficients is from 0.22 to 0.46 (mean : 0.40) for the daily precipitation sequences between the raw ERA-20c and weather stations over South Korea for the reference period (1973-2010). Thus, we used the ERA-20c data in the context of simulation data which is similar to the climate change scenarios during the reference period.

4. The authors proposed a new framework for bias correction in un-gauged area by using the IM-PCM method to interpolate the parameter of transfer functions. To my opinions, the contour mapping technique could bring large uncertainties and biases. Even the results are validated, but it is based on the average of all the 48 stations. The study of spatial impacts of this technique on the bias correction is still missing.

(**Response**) This study evaluated the IM-PCM method by employing a leave-one-out cross validation framework over 48 weather stations for the reference period and the overall error estimation results were described in the manuscript for both the extreme and mean. For a more specific analysis in each weather station in the context of cross validation, we generated a map showing the spatial errors in both annual maximum series (AMS) rainfalls and mean. The AMS errors were illustrated by root-mean-square-error (RMSE) and Nash-Sutcliffe efficiency (NSE) in Figure r1. For the mean, we further evaluated the IM-PCM method by estimating the relative error between the observed and modelled in Figure r2. As shown in the figures, for the AMS rainfalls, gpQM95 and gpQM99 generally perform well except for a few stations. Most stations showed NSE over 0.8 and RMSE less than 30mm. For the mean daily rainfall, the relative errors are generally below 10%.

[Figure]

*Figure r1. Cross validation results of the IM-PCM for the annual maximum series rainfall of the bias corrected data by QM approaches (gQM, gpQM95 and gpQM99) over 48 grid points. (a) Nash-Sutcliffe efficiency (NSE) and (b) root-mean-square-error (RMSE).*

[Figure]

*Figure r2.Relative error of the bias-corrected mean rainfall by QM approaches (gQM, gpQM95 and gpQM99) in 48 grid points compared with the corresponding in-situs.*

5. It is not clear that how the authors set the calibration and validation period. It seems the complete study period is used for both calibration and validation.

**(Response)** The main objective of this study is to correct the modelled data with the limited observation network. For this purpose, instead of temporally dividing the data into calibration and validation periods, we employed a leave-one-out cross validation scheme for the calibration and validation during the reference period to fully use the available data.

Specific comments:

Page 4, Line 5: "but not bias correction issues" -> "but not bias correction technique issues"

**(Response)** We have changed the sentence.

Page 4, Line 11: In addition to linear scaling, local intensity, power transformation, and quantile mapping, there are also there sophisticated method for bias correction, e.g. copula-based technique.

**(Response)** Thank you for the references. The literature review on the previous studies for bias correction has been changed as follows:
(Page 4, Line 9) "The underlying concepts for the bias correction approach vary from a simple delta change or mean bias correction to a quantile mapping (QM) or a multivariate approach based on the copula-based technique (Laux et al., 2011; Mao et al., 2015; Maraun, 2016; Maraun and Widmann, 2018; Teutschbein and Seibert, 2012).".

Page 5, Line 9: "Comparatively little attention has been given to the bias correction of the reanalysis data" is not correct. There is no clear clue for this.

(**Response**) There exist several studies on the bias correction of reanalysis, but it is rare to explore bias correction for century-long data such as ERA-20c. In this context, we have described that "Comparatively little attention has been given to the bias correction of the reanalysis data", but it needs to be described more specifically. Thus, we have changed the line to "Comparatively little attention has been given to the bias correction of the century-long reanalysis data like ERA-20c".

Page 10, Line 20: "(TH4). The frequency" -> "(TH4), the frequency"

(**Response**) Thank you for the comment. We have changed the typos.

Page 16, Line 9: It is not clear if all the TH are tested together with gQM or not?

(**Response**) All evaluations were based on the gQM along with the THs. We have clearly stated the explanation as follows:
(Page 16, Line 10) "This study examined four different thresholds (TH1, TH2, TH3, and TH4) for adjustment of the wet-day frequency of ERA-20c daily precipitation through an experiment with the gQM approach in terms of both the mean and extreme values".

Page 17, Line 13: "TH4 performs the best with 0.755 for NSE". It is a bit confuse to me, as the TH only affects the wet frequency of the time series. How it affects the extreme correction? This needs to be explained in more detail.

(**Response**) A cut-off threshold influences on the number of valid data in a given time series. To be more specific, the low threshold allows a relatively large number of data in the QM algorithm, while the high threshold reduces the number of valid data. Because the quantiles of extremes rely on the number of sample data in a given fitting curve, the threshold value affects the extreme correction based on the QM scheme. This explanation has been included in the revised manuscript.

Page 17, Line 20: It is not clear if the TH4 is take for gpQM tests.

(**Response**)  Thank you for the comment. Yes, gpQM approaches have been performed with TH4. We have clearly stated the explanation as follows:

(Page 18, Line 1) "Here, after adopting TH4 as a lower threshold, the 95th or 99th quantiles have been considered as upper thresholds for the correction of extremes (gpQM95 and gpQM99)".

[Reference ]

Fang, G., Yang, J., Chen, Y. N. and Zammit, C.: Comparing bias correction methods in downscaling meteorological variables for a hydrologic impact study in an arid area in China, Hydrol. Earth Syst. Sci., 19(6), 2547–2559, 2015.

Gutjahr, O. and Heinemann, G.: Comparing precipitation bias correction methods for high-resolution regional climate simulations using COSMO-CLM, Theor. Appl. Climatol., 114(3), 511–529, doi:10.1007/s00704-013-0834-z, 2013.

Laux, P., Vogl, S., Qiu, W., Knoche, H. R. and Kunstmann, H.: Copula-based statistical refinement of precipitation in RCM simulations over complex terrain, Hydrol. Earth Syst. Sci., 15(7), 2401–2419, doi:10.5194/hess-15-2401-2011, 2011.

Mao, G., Vogl, S., Laux, P., Wagner, S. and Kunstmann, H.: Stochastic bias correction of dynamically downscaled precipitation fields for Germany through Copula-based integration of gridded observation data, Hydrol. Earth Syst. Sci., 19(4), 1787–1806, doi:10.5194/hess-19-1787-2015, 2015.

Maraun, D.: Bias Correcting Climate Change Simulations - a Critical Review, Curr. Clim. Chang. Reports, 2(4), 211–220, doi:10.1007/s40641-016-0050-x, 2016.

Maraun, D. and Widmann, M.: Statistical Downscaling and Bias Correction for Climate Research, Cambridge University Press., 2018.

Nyunt, C. T., Koike, T. and Yamamoto, A.: Statistical bias correction for climate change impact on the basin scale precipitation in Sri Lanka , Philippines , Japan and Tunisia, , (January), doi:10.5194/hess-2016-14, 2016.

Poli, P., Hersbach, H., Tan, D., Dee, D., Thépaut, J.-N., Simmons, A., Peubey, C., Laloyaux, P., Komori, T., Berrisford, P., Dragani, R., Trémolet, Y., Holm, E., Bonavita, M., Isaksen, L. and Fisher, M.: The data assimilation system and initial performance evaluation of the ECMWF pilot reanalysis of the 20th-century assimilating surface observations only (ERA-20C)., 2013.

Smith, A., Freer, J., Bates, P. and Sampson, C.: Comparing ensemble projections of flooding against flood estimation by continuous simulation, J. Hydrol., 511, 205–219, doi:10.1016/j.jhydrol.2014.01.045, 2014.

Teng, J., Potter, N. J., Chiew, F. H. S., Zhang, L., Wang, B., Vaze, J. and Evans, J. P.: How does bias correction of regional climate model precipitation affect modelled runoff ?, , 711–728, doi:10.5194/hess-19-711-2015, 2015.

Teutschbein, C. and Seibert, J.: Bias correction of regional climate model simulations for hydrological climate-change impact studies: Review and evaluation of different methods, J. Hydrol., 456, 12–29, 2012.

Themeßl, M. J., Gobiet, A. and Leuprecht, A.: Empirical-statistical downscaling and error correction of daily precipitation from regional climate models, Int. J. Climatol., 31(10), 1530–1544, doi:10.1002/joc.2168, 2011.

Volosciuk, C., Maraun, D., Vrac, M. and Widmann, M.: A combined statistical bias correction and stochastic downscaling method for precipitation, Hydrol. Earth Syst. Sci., 21(3), 1693–1719, doi:10.5194/hess-21-1693-2017, 2017.

Vrac, M. and Naveau, P.: Stochastic downscaling of precipitation : From dry events to heavy rainfalls, , 43(July), 1–13, doi:10.1029/2006WR005308, 2007.

---

## Author Comment (AC2) · 11 May 2018

**Authors' response to Referee #2**

For clarity, authors' responses are presented by blue colour.

We have answered all the comments of the reviewer 2. Answers are attached to this revision note. Along with the answers we are also explaining all the changes we have done.

Review of "Exploring the Long-Term Reanalysis of Precipitation and the Contribution of Bias Correction to the Reduction of Uncertainty over South Korea: A Composite Gamma-Pareto Distribution Approach to the Bias Correction" by Kim et al.

The authors present and evaluate a bias correction of the ECMWF ERA-20c reanalysis for South Korea. The correction is based on a parametric quantile mapping and calibrated between reanalysis grid-box and observed station precipitation, and extended to the full field by interpolating the transfer function parameters in space.

I cannot recommend this manuscript for publication. At least major parts should be substantially revised, and the spatial model should be fully omitted. My major concerns are as follows:

1. Deterministic bias correction of precipitation cannot be used for downscaling, and in particular not to create spatial fields. Maraun (2013) has demonstrated that bias correction suffers from the same conceptual flaw as inflated regression. Differences between reanalysis and station observations (in particular the magnitudes of summer extremes and wet day frequencies) are not necessarily biases, but to a substantial degree due to the scale gap between the area average of the reanalysis and the pointscale of the observations. Local-scale variability is not fully determined by the grid box average, a deterministic rescaling as done by quantile mapping cannot create the missing local variability. Instead the large-scale variability is inflated. Thus, the corrected time series have similar marginal properties as the local observations, but do not have the correct spatial-temporal properties. This is a problem in particular for spatial fields, as the spatial distribution of the corrected field is still that of the reanalysis (apart from the wet-day correction), but only inflated. It does not represent the smallscale variability of summer thunderstorms, e.g. The problem is severe for extreme events: dry areas as well as the magnitude of precipitation falling over a certain area are substantially overestimated (Maraun 2013). Thus, using these data for hydrological modeling would likely result in dangerously misleading results. This issue is rather irritating, given that the authors cite Volosciuk et al. (2017) who discuss this issue in depth. In fact, the only correct solution would be a stochastic bias correction that bias corrects (if needed at all in this case) and

additionally adds random small-scale variability (either in a single-site approach as suggested by Volosciuk et al. (2013) or with a fully spatial model. If only single locations are considered (without using time series at multiple sites), a quantile mapping to the point-scale might be justified based on pragmatic reasoning.

A major problem of the manuscript is that the evaluation is essentially blind to these problems. They are mostly visible in the spatial and interannual variability. None of these aspects have been evaluated.

(**Response**) Thank you for the comments and the explanations. We agree that there may exist the spatial bias between local station observation and gridded reanalysis, and the bias corrected values could misrepresent spatial-temporal pattern. However, a primary objective of this study is to statistically extend the sample size, especially for extreme values, in a certain area with spatio-temporally sparse observation network. Therefore, specific day-to-day variation or trend analysis was not our main concern in this study. In these perspectives, we further investigated the IM-PCM method within a leave-one-out cross validation framework.

First, we evaluated the IM-PCM method by employing a leave-one-out cross validation framework over 48 weather stations for the reference period and the overall error estimation results were described in the manuscript for both the extreme and mean. For a more specific analysis in each weather station in the context of cross validation, we generated a map showing the spatial errors in both annual maximum series (AMS) rainfalls and mean. The AMS errors were illustrated by root-mean-square-error (RMSE) and Nash-Sutcliffe efficiency (NSE) in Figure S1. For the mean, we further evaluated the IM-PCM method by estimating the relative error between the observed and modelled in Figure S2. As shown in the figures, for the AMS rainfalls, gpQM95 and gpQM99 generally perform well except for a few stations. Most stations showed NSE over 0.8 and RMSE less than 30mm. For the mean daily rainfall, the relative errors are generally below 10%.

*(a)*

[Figure]

[Figure]

*Figure S1. Cross validation results of the IM-PCM for the annual maximum series rainfall of the bias corrected data by QM approaches (gQM, gpQM95 and gpQM99) over 48 grid points. (a) Nash-Sutcliffe efficiency (NSE) and (b) root-mean-square-error (RMSE).*

[Figure]

*Figure S2.Relative error of the bias-corrected mean rainfall by QM approaches (gQM, gpQM95 and gpQM99) in 48 grid points compared with the corresponding in-situs.*

Second, one generally collects annual maximum rainfall series or extreme values over a certain threshold from historical records to derive Intensity-Duration-Frequency (IDF) relationships. In many regions including South Korea, the long-term meteorological record for a given catchment is largely limited to evaluate the reliable IDF relationships. Thus, the uncertainty of the estimated design rainfalls could be affected by uncertainty associated with the sampling error (Coles et al., 2003; Huard et al., 2010; Overeem et al., 2008; Tung and Wong, 2014; Van de Vyver, 2015). We further explored the uncertainty range of design rainfalls based on GEV distribution for a given return period and a given data length within a Bayesian modelling framework. As illustrated in Figure S3, the uncertainty range of design rainfall is significantly reduced with increasing data.

[Figure]

*Figure S3. Boxplot for the uncertainty range of design rainfalls with 30-yr, 50-yr and 100-yr return period based on GEV distribution for 38 annual maximum series (AMS) (Data(38)) and 111 AMS data (Data(111).*

2. The discussion in the manuscript is rather naive and largely ignores problems of bias correction and reanalysis data. It also ignores much of the literature in the field. For instance, it is well known that at least the first versions of century-long reanalysis data strongly misrepresent long-term climatic trends, or that synoptic-scale variability in the Tropics is only weakly constraint in reanalysis data (Krueger et al, 2013; Befort et al., 2016; Brands et al., 2012). These issues are not discussed in the manuscript. Similarly, the downscaling issues discussed above have not been acknowledged, differences between biases and scale-gaps in the given example have not been discussed. In fact, the authors do not make any attempt to discuss which kind of biases can be corrected in their context. E.g., misrepresented long-term trends, spatial-temporal variability (apart from wet-day corrections) or a misrepresented tropical day-to-day variability will not be corrected by the bias correction. See, e.g., Maraun et al., 2017, for a discussion of several issues (many are relevant in a climate change context, but some apply also here).

(**Response**) Thank you for the valuable comments. As aforementioned above, this study aims to explore a composite distribution based bias correction approach for century-long ERA-20c daily precipitation data which could be possibly used to reduce the uncertainty associated with sampling error in rainfall frequency analysis. Thus, other issues related to the bias correction for daily rainfall series have not been fully considered. We agree that those discussions are valuable for this study, and the discussion has been included in the revised manuscript as follows:

"(Page 3, Line 19) However, although substantial improvements have been made in the modelling process, previous studies have shown that reanalysis datasets still have their own systematic errors which vary in space and time (Bao and Zhang, 2013; Bosilovich et al., 2008; Gao et al., 2016; Kim and Han, 2018; Ma et al., 2009) It is also clear that century-long reanalysis data may misrepresent long-term climatic trends or synoptic scale variability, especially for the first half of twentieth century, and there exists the difference in temporal variability between century-long reanalyses (Befort et al., 2016; Brands et al., 2012; Donat et al., 2016; Krueger et al., 2013; Poli et al., 2013)."

"(Page 21, Line 17) One major issue in QM approach is that the bias corrected values could suffer from inflation of the bias corrected value or implausible representation of temporal pattern (Bum Kim et al., 2016; Maraun, 2013, 2016; Maraun et al., 2017; Volosciuk et al., 2017; Vrac and Friederichs, 2015). However, as this study aims to statistically extend the sample size, especially for extreme values associated with the sampling error in rainfall frequency analysis, in a certain area with spatio-temporally sparse observation network, specific day-to-day variation or trend analysis has not been fully considered. One generally derives Intensity-Duration-Frequency (IDF) relationships for rainfall intensity analysis, but in many regions including South Korea, the long-term meteorological record for a given catchment is largely limited. Thus, the uncertainty of the estimated design rainfalls could be affected by uncertainty associated with the sampling error (Coles et al., 2003; Huard et al., 2010; Overeem et al., 2008; Tung and Wong, 2014; Van de Vyver, 2015). I.e., the more the reliable sample data, the less uncertainty exists in the estimated rainfall depth. In these perspectives, the bias correction methods developed in this study both statistically improved the quality of the data and extended daily precipitation over the 20th century in South Korea."

3. The language needs substantial revision, as well as the logic within several sentences. I will give some examples below.

(**Response**) We have carefully revised the manuscript. The English structure and grammar of the manuscript has been thoroughly reviewed through a specialized English editing office for proof reading. Thanks for the constructive comments again.

Further comments:
p2|11: this sentence makes no sense and does not logically link to the previous sentence.

(**Response**) In this paragraph, we first described the influence of climate change on a wide range of fields and moved the focus into the impact on water related hazards. We have modified this paragraph as follows:

"Recent studies have documented that long-term climate change has influenced a wide range of fields such as agriculture, environment, health, economy and water resources (IPCC, 2014; Nelson et al., 2009; Patz et al., 2005; Vörösmarty et al., 2000). An increase or decrease in climate variables such as precipitation and temperature can affect the growth of crops, ecosystem, human diseases, and water-related hazards such as floods and droughts. Of these impacts, water-related hazards are closely linked to changes in rainfall intensity, which are of primary concern to water resource managers.

p2|16: the data are not just coarsely represented in model calibration, they are simply coarse.

(**Response**) This sentence has been changed as follows:
"However, it has been widely acknowledged that the observed data are coarse in space, and long-term climate data are not readily available in many countries around the world."

p2|23: what does "finer" refer to? Or should it be just "fine"? In any case I would not agree that reanalysis are provided at a fine resolution. What is more important is that they provide a complete field.

(**Response**) The term 'finer' was used to emphasize the spatial aspects of reanalysis data compared with observation, because the reanalysis can provide spatial information for a variety of climate variables. To be clearer, this sentence has been changed as follows:
"A primary strength of the reanalysis data is that compared with observation, they provide spatial information with a longer period, a few of which can cover the whole 20th century."

p3|2: "spans from" English!

(**Response**) "spans from" has been changed into "spanning from".

p3|11-13: this does not make sense. If pressure and wind are not assimilated, how can the synoptic situation then be represented?

(**Response**) For ERA-20cm, no atmospheric observations were assimilated (Hersbach et al., 2015). For this reason, ERA-20cm is not able to reproduce actual synoptic situations as described in the manuscript, but the ensemble can provide a statistical estimate of the climate. The detailed on ERA-20cm is found in Hersbach et al. (2015).

p3|14: what does "on the other hand" refer to?

(**Response**) This was used to distinguish descriptions for NOAA-20cR, which followed after this phrase, from the products from the ECMWF (ERA-20c and ERA-20cm) described before this phrase.

p3: here the limitations of the reanalysis data should be discussed.

(**Response**) As aforementioned above, this study aims to statistically extend the sample size, especially for extreme values, in a certain area with spatio-temporally sparse observation network. In this vein, we have not fully described the limitations of the reanalysis. We agree that those discussions are valuable for this study, so the limitations has been included in the revised manuscript as follows:
"(Page 3, Line 19) However, although substantial improvements have been made in the modeling process, previous studies have shown that reanalysis datasets still have their own systematic errors which vary in space and time (Bao and Zhang, 2013; Bosilovich et al., 2008; Gao et al., 2016; Kim and Han, 2018; Ma et al., 2009) It is also clear that century-long reanalysis data may misrepresent long-term climatic trends or synoptic scale variability, especially for the first half of twentieth century, and there exists the difference in temporal variability between century-long reanalyses (Befort et al., 2016; Brands et al., 2012; Donat et al., 2016; Krueger et al., 2013; Poli et al., 2013)".

p4|9: there is a more recent review by Maraun (2016) and the recent book by Maraun and Widmann (2018). Also the selection of methods is rather arbitrary.

(**Response**) Thank you for the references. The literature review on the previous studies for bias correction has been included as follows:
"The underlying concepts for the bias correction approach vary from a simple delta change or mean bias correction to a quantile mapping (QM) or a multivariate approach based on copulabased technique (Laux et al., 2011; Mao et al., 2015; Maraun, 2016; Maraun and Widmann, 2018; Teutschbein and Seibert, 2012)."

p4|13: bias correction cannot reduce errors in numerical models! It can, at best, postprocess numerical models.

(**Response**)  We have changed the sentences as follows:
"Although each method has its own merits and limitations, previous studies have shown that bias correction methods were generally capable of reducing systematic errors of numerical model outputs and, among them, QM showed better performance than other approaches, especially for precipitation."

p4|14: "Jacob Themessl et al" should be "Themessl et al.". The name is Matthias J. Themessl.

(**Response**) We have changed the name of the citation.

 p4|15 "referred to as other names" grammar!

(**Response**) We have changed "referred to as other names such as 'distribution mapping' and 'probability mapping' " to "referred to as 'distribution mapping' or 'probability mapping' ".

p4|18 "usually based on a gamma". No - this is not true. There are many other implementations, and often non-parametric approaches are used.

(**Response**) We agree that there are various transfer functions from empirical CDF to a composite distribution in quantile mapping for precipitation. As the proposed methodology was contrived to interpolate transfer function parameters, we did not explore the non-parametric approach in this study. Among parametric QMs, the gamma distribution is routinely adopted for monotonous fitting, while current studies have employed a composite or mixture distribution for improving extreme values more effectively as described in the manuscript. In this context, we have changed the sentence as follows:
"The QM method, referred to as 'distribution mapping' or 'probability mapping', was used to rectify the cumulative distribution of the modelled data against that of the observed data by employing a transfer function. To allow for interpolation in space and extrapolation in time,

the parametric QM-approach is considered in this study. For daily precipitation, a gamma distribution is commonly adopted in parametric fitting."

p5|1: "underestimation" Not necessarily. In particular moderate extremes might be overestimated (in the range where the scale parameter dominates).

(**Response**) We have changed the sentence as follows:
"In other words, the gQM approach may result in misrepresentation of the upper tail of the distribution, which, in turn, can lead to underestimation of the design rainfalls."

p5|13 and following: as discussed above, this approach is not sensible, at least not for a deterministic method which is interpreted at multiple sites.

(**Response**) As aforementioned in response to 1., a primary objective of this study is to statistically extend the sample size, especially for extreme values, in a certain area with spatio-temporally sparse observation network. Although there still exist some biases in the IM-PCM method, the uncertainty range of design rainfall might be significantly reduced with increasing data. In these perspectives, the suggested approach is still meaningful.

p12|12-16: this listing is a bit naive. The GEV is designed to model block maxima. It may fit a distribution tail rather well because it is flexible (3 parameters), but conceptually this doesn't make sense. Here some discussion should be added.

(**Response**) To identify the best marginal distribution, we have considered GPD, GEV, Gumbel, Weibull and Lognormal which have been commonly adopted in rainfall frequency analysis. As you indicated, we agree that the use of GEV distribution cannot be justified for the composite distribution. Thus, we removed GEV distribution. Thanks for the comments.

p13, eq. (3): this model is a bit crude. There are many implementations that ensure at least continuity at the transition point between gamma and GPD, some even smooth-ness. The method here essentially has a jump.

(**Response**) Thanks for the comments. We pragmatically adopted a composite of two distributions because the non-continuity over the threshold does not affect the overall bias results.

p14|2: "mainly" well, what other reason should there be?

(**Response**) The term "mainly" has been omitted.

Section 3.3: as discussed, this is extremely dangerous and should not be done.

(**Response**)

As aforementioned in the response to 1., a primary objective of this study is to statistically extend the sample size, especially for extreme values, in a certain area with spatio-temporally sparse observation network. Although there still exist some biases in the IM-PCM method, the uncertainty range of design rainfall could be significantly reduced with increasing data. In these perspectives, the suggested approach is still meaningful.

[References]
Befort, D. J., Wild, S., Kruschke, T., Ulbrich, U. and Leckebusch, G. C. (2016), 'Different long- term trends of extra-tropical cyclones and windstorms in ERA-20C and NOAA-20CR reanalyses', Atmos. Sci. Lett. 17(11), 586–595.
Brands, S., Gutiérrez, J. M. and Herrera, S. (2012), 'On the use of reanalysis data for downscaling', J. Climate 25, 2517–2526.
Krueger, O., Schenk, F., Feser, F. and Weisse, R. (2013), 'Inconsistencies between long- term trends in storminess derived from the 20CR reanalysis and observations', J. Climate 26(3), 868–874.
Maraun, D. (2013), 'Bias correction, quantile mapping and downscaling: revisiting the inflation issue', J. Climate 26, 2137–2143.
Maraun, D. (2016), 'Bias correcting climate change simulations - a critical review', Curr. Clim. Change Rep. 2(4), 211–220.
Maraun, D., Shepherd, T. G., Widmann, M., Zappa, G., Walton, D., Gutierrez, J. M., Hage- mann, S., Richter, I., Soares, P. M. M., Hall, A. and Mearns, L. (2017b), 'Towards process-informed bias correction of climate change simulations', Nat. Clim. Change, online first, DOI 10.1038/nclimate3418.
Maraun, D., and Widmann, M. (2018), 'Statistical Downscaling and Bias correction for Climate Research ', Cambridge University Press.
Volosciuk, C., Maraun, D., Vrac, M. and Widmann, M. (2017), 'A combined statistical bias correction and stochastic downscaling method for precipitation', Hydrol. Earth Syst. Sci. 21(3), 1693–1719.

[Reference]

Bao, X. and Zhang, F.: Evaluation of NCEP–CFSR, NCEP–NCAR, ERA-Interim, and ERA-40

reanalysis datasets against independent sounding observations over the Tibetan Plateau, J. Clim., 26(1), 206–214, 2013.

Befort, D. J., Wild, S., Kruschke, T., Ulbrich, U. and Leckebusch, G. C.: Different long-term trends of extra-tropical cyclones and windstorms in ERA-20C and NOAA-20CR reanalyses, Atmos. Sci. Lett., 17(11), 586–595, doi:10.1002/asl.694, 2016.

Bosilovich, M. G., Chen, J., Robertson, F. R. and Adler, R. F.: Evaluation of global precipitation in reanalyses, J. Appl. Meteorol. Climatol., 47(9), 2279–2299, 2008.

Brands, S., Gutiérrez, J. M., Herrera, S. and Cofiño, A. S.: On the use of reanalysis data for downscaling, J. Clim., 25(7), 2517–2526, doi:10.1175/JCLI-D-11-00251.1, 2012.

Coles, S., Pericchi, L. R. and Sisson, S.: A fully probabilistic approach to extreme rainfall modeling, J. Hydrol., 273(1–4), 35–50, doi:10.1016/S0022-1694(02)00353-0, 2003.

Donat, M. G., Alexander, L. V, Herold, N. and Dittus, A. J.: Temperature and precipitation extremes in century-long gridded observations, reanalyses, and atmospheric model simulations, J. Geophys. Res. Atmos., 121(19), 2016.

Fang, G., Yang, J., Chen, Y. N. and Zammit, C.: Comparing bias correction methods in downscaling meteorological variables for a hydrologic impact study in an arid area in China, Hydrol. Earth Syst. Sci., 19(6), 2547–2559, 2015.

Gao, L., Bernhardt, M., Schulz, K., Chen, X. W., Chen, Y. and Liu, M. B.: A First Evaluation of ERA-20CM over China, Mon. Weather Rev., 144(1), 45–57, doi:10.1175/Mwr-D-15-0195.1, 2016.

Hersbach, H., Peubey, C., Simmons, A., Berrisford, P., Poli, P. and Dee, D.: ERA-20CM: a twentieth-century atmospheric model ensemble, Q. J. R. Meteorol. Soc., 141(691), 2350–2375, 2015.

Huard, D., Mailhot, A. and Duchesne, S.: Bayesian estimation of intensity-duration-frequency curves and of the return period associated to a given rainfall event, Stoch. Environ. Res. Risk Assess., 24(3), 337–347, doi:10.1007/s00477-009-0323-1, 2010.

IPCC: Climate Change 2014–Impacts, Adaptation and Vulnerability: Regional Aspects, Cambridge University Press., 2014.

Kim, D.-I. and Han, D.: Comparative study on long term climate data sources over South Korea, J. Water Clim. Chang. [online] Available from: http://jwcc.iwaponline.com/content/early/2018/03/14/wcc.2018.032.abstract, 2018.

Krueger, O., Schenk, F., Feser, F. and Weisse, R.: Inconsistencies between long-term trends in storminess derived from the 20CR reanalysis and observations, J. Clim., 26(3), 868–874, doi:10.1175/JCLI-D-12-00309.1, 2013.

Laux, P., Vogl, S., Qiu, W., Knoche, H. R. and Kunstmann, H.: Copula-based statistical refinement of precipitation in RCM simulations over complex terrain, Hydrol. Earth Syst. Sci., 15(7), 2401–2419, doi:10.5194/hess-15-2401-2011, 2011.

Ma, L., Zhang, T., Frauenfeld, O. W., Ye, B., Yang, D. and Qin, D.: Evaluation of precipitation from the ERA-40, NCEP-1, and NCEP-2 Reanalyses and CMAP-1, CMAP-2, and GPCP-2 with ground-based measurements in China, J. Geophys. Res. Atmos., 114(D9), 2009.

Mao, G., Vogl, S., Laux, P., Wagner, S. and Kunstmann, H.: Stochastic bias correction of dynamically downscaled precipitation fields for Germany through Copula-based integration of gridded observation data, Hydrol. Earth Syst. Sci., 19(4), 1787–1806, doi:10.5194/hess-19-1787-2015, 2015.

Maraun, D.: Bias Correcting Climate Change Simulations - a Critical Review, Curr. Clim. Chang. Reports, 2(4), 211–220, doi:10.1007/s40641-016-0050-x, 2016.

Maraun, D. and Widmann, M.: Statistical Downscaling and Bias Correction for Climate Research, Cambridge University Press., 2018.

Nelson, G. C., Rosegrant, M. W., Koo, J., Robertson, R., Sulser, T., Zhu, T., Ringler, C., Msangi, S., Palazzo, A. and Batka, M.: Climate change: Impact on agriculture and costs of adaptation, Intl Food Policy Res Inst., 2009.

Overeem, A., Buishand, A. and Holleman, I.: Rainfall depth-duration-frequency curves and their uncertainties, J. Hydrol., 348(1–2), 124–134, doi:10.1016/j.jhydrol.2007.09.044, 2008.

Patz, J. A., Campbell-Lendrum, D., Holloway, T. and Foley, J. A.: Impact of regional climate change on human health, Nature, 438(7066), 310–317, 2005.

Poli, P., Hersbach, H., Tan, D., Dee, D., Thépaut, J.-N., Simmons, A., Peubey, C., Laloyaux, P., Komori, T., Berrisford, P., Dragani, R., Trémolet, Y., Holm, E., Bonavita, M., Isaksen, L. and Fisher, M.: The data assimilation system and initial performance evaluation of the ECMWF pilot reanalysis of the

20th-century assimilating surface observations only (ERA-20C)., 2013.

Teutschbein, C. and Seibert, J.: Bias correction of regional climate model simulations for hydrological climate-change impact studies: Review and evaluation of different methods, J. Hydrol., 456, 12–29, 2012.

Themeßl, M. J., Gobiet, A. and Leuprecht, A.: Empirical-statistical downscaling and error correction of daily precipitation from regional climate models, Int. J. Climatol., 31(10), 1530–1544, doi:10.1002/joc.2168, 2011.

Tung, Y. koung and Wong, C. leung: Assessment of design rainfall uncertainty for hydrologic engineering applications in Hong Kong, Stoch. Environ. Res. Risk Assess., 28(3), 583–592, doi:10.1007/s00477-013-0774-2, 2014.

Vörösmarty, C. J., Green, P., Salisbury, J. and Lammers, R. B.: Global water resources: vulnerability from climate change and population growth, Science (80-. )., 289(5477), 284–288, 2000.

Van de Vyver, H.: Bayesian estimation of rainfall intensity-duration-frequency relationships, J. Hydrol., 529, 1451–1463, doi:10.1016/j.jhydrol.2015.08.036, 2015.

---

## Editor Comment (EC1) · B. Schaefli (Editor) · 14 May 2018

To give a preliminary summary of the reviews, I would like to point out here that the first two reviewers are both very critical about the scientific significance and the scientific quality of this manuscript. Reviewer 1 e.g. states that "the contribution of this paper to the scientific progress is low.". And even if the overall comments of reviewer 1 seem rather positive, this reviewer, in the formal paper evaluation to the editor, recommends

rejection of this manuscript.

The critics of reviewer 2, who also recommends rejection, are more profound. Reviewer 2 states and discusses that i) "deterministic bias correction of precipitation cannot be used for downscaling, and in particular not to create spatial fields" and ii) "The discussion in the manuscript is rather naive and largely ignores problems of bias correction and reanalysis data. It also ignores much of the literature in the field. (..) the authors do not make any attempt to discuss which kind of biases can be corrected in their context.".

The authors have provided answers to the first two reviews. Before I give a final comment on how to proceed with the paper, based also on these answers, I am waiting for a third review.

---

## Referee Comment (RC3) · Anonymous Referee #3 · 18 May 2018

The goal of this study is to present and evaluate a bias correction of the ECMWF ERA-20c reanalysis for South Korea. The authors apply a combination of transfer functions and wet frequency adjustment methods to correct the bias present in the precipitation time series. Parameters of the obtained transfer functions derived from the relation between the reanalysis grid and observed rain gauge precipitation are interpolated in space to full grid precipitation data. Overall evaluation: This is a potentially interesting paper, but in order to be published a major revision is required. The results presented

in this paper are relatively simple and lack of deep analysis. The authors provide a long text on motivation for bias correction but omit the discussion of the bias correction in context of downscaling and do not discuss the constrains and limitations of the parameter interpolation. The authors claim in the title reduction of uncertainty but do not prove that this is the case.

1. Overall I am left unclear on the core contribution of the paper. The evaluation of the ERA precipitation over South Korea is a valuable contribution but it is very short. The applied bias correction is described in detail but a justification is missing. Finally, the spatial interpolation is not correctly validated.

2. Reviewers #1 and #2 provide excellent recommendations and there is no need to repeat them here. Along the lines outlined there the manuscript can be improved.

The authors may consider to rename Section 2 to "Material and methods" and to describe in addition to sections 2.1 and 2.2 in two new sections 2.3. and 2.4 the BC and the downscaling issues, possibly with some text from the introduction in which the scientific goals of the study should be clearly identified. Based on the findings and constrains discussed in this section the applied methodology can be justified and presented in detail in section 3. "Applied methodology". The validation procedure should include an analysis and discussion of the differences between the calculated and observed values at each station when this station is not included into the derivation of the interpolated parameters. This will help to access the possible errors at ungauged grid cells and thus help to judge the entire applied procedure and draw correct conclusions.

3. As a minimum requirement before revision, the manuscript has to be professionally revised and edited to correct the language and to remove the unnecessary text repetitions

Specific:

p. 9 l. 1 - What is the rationale for using stations 4,16,28, 40 ?

- Large deviations are also visible in spring

p.9 l. 3 - The bias extreme is proportional . . . I cannot see this, and even would argue that the maximum rain at station 4 is a mistake in station reading

p.9 l. 6 -This paragraph is supposed to summarize the section 2.2, but after the summary it introduces a new investigated item: wet-day. This should be presented after line 5 on page 3. Also, I suggest to add a short description of the applied evaluation statistics after the introduction of ERA-20c.

p.9 l. 9 What is the role of climate models here ?

p.9. l. 12 Explain wet-day (RR > 1 mm/d ?)

p. 3 l. 21 This paragraph is an unnecessary repetition of the summary in the last paragraph.

p.12 l. 12-20 Some explanation of AIC and BIC and discussion why DGP was chosen is needed here. I cannot understand the title and the content of Table 2.

p. 13 l. 5 "Again, . . ." repeats line 3

p. 15 l. 5 "I . . . the suitability . . ." for what? This goal of the study has not been mentioned I the introduction.

p. 15 l. 19 ". . . leave-one-out procedure.. " The procedure definitely needs a longer explanation and discussion. Usually one period is used for training and an another for validation.

p. 16 l. 6 Where is section 3.4.1 ?. I my opinion the section "Evaluation criteria "should be in section 2. Material and methods

p. 17 l. 1 As illustrated in the previous section. . . The range 0.-4.66 is not mentioned in the previous section.

p. 17 l. 5 What is "the degree of bias" ?

p. 17 l. 6 "... significantly varied..." add some numbers here to quantify this variation

p.17 l. 21. "This study introduces ..." rewrite to This study applies

p. 19, l. 10-13 " In other words.. " This is trivial. If there is no difference between model and observation then there is no need for a bias correction

p. 21 l. 16 "The bias correction . . . improved the quality . . ." Perhaps the mean over the region. What can be said for ungauged regions?

Figure 1. Indicate the location of the gauges 4,16,28, and 40 used in evaluation

---

## Author Comment (AC3) · 30 May 2018

**Authors' response to Referee #3**

For clarity, authors' responses are presented by blue colour.

We have answered all the comments of the reviewer 3. Answers are attached to this revision note. Along with the answers we are also explaining all the changes we have done.

The goal of this study is to present and evaluate a bias correction of the ECMWF ERA-20c reanalysis for South Korea. The authors apply a combination of transfer functions and wet frequency adjustment methods to correct the bias present in the precipitation time series. Parameters of the obtained transfer functions derived from the relation between the reanalysis grid and observed rain gauge precipitation are interpolated in space to full grid precipitation data. Overall evaluation: This is a potentially interesting paper, but in order to be published a major revision is required. The results presented in this paper are relatively simple and lack of deep analysis. The authors provide a long text on motivation for bias correction but omit the discussion of the bias correction in context of downscaling and do not discuss the constrains and limitations of the parameter interpolation. The authors claim in the title reduction of uncertainty but do not prove that this is the case.

1. Overall I am left unclear on the core contribution of the paper. The evaluation of the ERA precipitation over South Korea is a valuable contribution but it is very short. The applied bias correction is described in detail but a justification is missing. Finally, the spatial interpolation is not correctly validated.

(**Response**) Thank you for the constructive comments. After the preliminary evaluation of the ERA-20c daily precipitation over South Korea, this study mainly focused on the bias correction of ERA-20c daily precipitation, especially for extreme values, because the century-long precipitation dataset could contribute to the reduction of the uncertainty in hydrologic frequency analysis where a limited number of observations were generally given. As indicated, the bias correction is generally involved with downscaling of general circulation models (GCMs). More specifically, the spatial resolutions of GCMs are too coarse to adequately represent regional climate variability so that the direct use of those model outputs is not appropriate, especially for fine-scale hydrological applications. Moreover, most model outputs in climate models are affected by spatio-temporal biases, leading to significant bias in hydrological impact studies. In these contexts, both bias correction and spatial downscaling of model outputs are crucially involved in the use of GCMs for hydrological

impact studies. In our case, spatial resolution of ERA-20c (i.e.  $0.125^{\circ} \times 0.125^{\circ}$ ) is relatively high enough to be used in practical applications. Therefore, the spatial downscaling has not been considered in the current study and we rather much focused on certain aspects of the bias correction, which might be of importance for a special use of the long-term reanalysis data in hydrologic frequency analysis (Coles et al., 2003; Huard et al., 2010; Overeem et al., 2008; Tung and Wong, 2014; Van de Vyver, 2015). In order to validate the use of long-term reanalysis data for the reduction of uncertainty in estimating design rainfalls, we further explored the uncertainty range of design rainfalls based on GEV distribution for a given return period and a given data length within a Bayesian modelling framework. As illustrated in Figure A1, the uncertainty range of design rainfall is significantly reduced with increasing data. In this perspective, we applied the suggested QM approach in this study.

Figure A1. Boxplot for the uncertainty range of design rainfalls with 30-yr, 50-yr and 100-yr return period based on GEV distribution for 38 annual maximum series (AMS) (Data(38)) and 111 AMS data (Data(111).

For spatial interpolation of a set of parameters associated with transfer functions in quantile mapping approach, this study further evaluated the IM-PCM method by employing a leave-one-out cross validation framework over 48 weather stations for the reference period (1973-2010) and the overall performance has been illustrated in the manuscript for both the extreme and mean. For a more specific analysis in each weather station in the context of cross validation, we generated a map showing the spatial errors in both annual maximum series (AMS) rainfalls and mean. The AMS errors were evaluated by root-mean-square-error (RMSE) and Nash-Sutcliffe efficiency (NSE) in Figure A2. For the mean, we additionally evaluated the IM-PCM method by estimating the relative error between the observed and modelled in Figure A3. As shown in the figures, for the AMS rainfalls, gpQM95 and

gpQM99 generally perform well except for a few stations. Most stations showed NSE over 0.8 and RMSE less than 30mm. For the mean daily rainfall, the relative errors are generally below 10%.

---

## Editor Comment (EC2) · B. Schaefli (Editor) · 4 Jun 2018

As summarized by reviewer 2, "The authors present and evaluate a bias correction of the ECMWF ERA-20c reanalysis for South Korea. The correction is based on a parametric quantile mapping and calibrated between reanalysis grid-box and observed station precipitation, and extended to the full field by interpolating the transfer function parameters in space."

The context of the underlying study was however not clear. As formulated in the response to reviewer 3, "this study mainly focused on the bias correction of ERA-20c daily precipitation, especially for extreme values, because the century-long precipitation dataset could contribute to the reduction of the uncertainty in hydrologic frequency analysis where a limited number of observations were generally given.". In other words, this study proposes to apply bias correction to a long precipitation reanalysis data set to improve the estimation of design variables. In response to reviewer 2, this is formulated as "However, a primary objective of this study is to statistically extend the sample size, especially for extreme values, in a certain area with spatio-temporally sparse observation network." With this respect, the title and the abstract where misleading ("Exploring the Long-Term Reanalysis of Precipitation and the Contribution of Bias Correction to the Reduction of Uncertainty over South Korea: A Composite Gamma-Pareto Distribution Approach to the Bias Correction").

The three reviewer were very critical about the manuscript both in terms of scientific quality and relevance. Two reviewers suggested rejection, one major revisions.

Reviewer 3 questions the novelty of the discussed methods. In their response, the authors argue that the novelty lies in the fact that the presented methods are applied to "long reanalysis data" (rather than short reanalysis data sets), which then opens new perspectives for extreme event analysis.

While this might a prior justify publication of this manuscript, reviewer 2 discusses in detail why the proposed bais correction method is not useful for extreme event analysis. In summary (cited from reviewer 2) "Deterministic bias correction of precipitation cannot be used for downscaling, and in particular not to create spatial fields. (..), the corrected time series have similar marginal properties as the local observations, but do not have the correct spatial-temporal properties. This is a problem in particular for spatial fields, as the spatial distribution of the corrected field is still that of the reanalysis (apart from the wet-day correction), but only inflated."
The authors response to this comment is not satisfying ("We agree that there may exist the spatial bias between local station observation and gridded reanalysis, and the bias corrected values could misrepresent spatial-temporal pattern. However, a primary objective of this study is to statistically extend the sample size, especially for extreme values, in a certain area with spatio-temporally sparse observation network. Therefore, specific day-to-day variation or trend analysis was not our main concern in this study."). The authors do in particular not discuss the problem of spatial rainfall properties. Their reasoning is motivated by how design rainfall is usually obtained ("Second, one generally collects annual maximum rainfall series or extreme values over a certain threshold from historical records to derive Intensity-Duration-Frequency (IDF) relationships."). From an engineering practice viewpoint, it might indeed seem a possible option to extend rainfall records at individual stations without accounting explicitly for the properties of the spatial rainfall field. From scientific viewpoint, however, it is highly questionable to produce time series at individual stations without accounting for spatial properties.

The authors furthermore justify their approach by the uncertainty reduction obtained if longer time series are used for design rainfall estimation (extreme event analysis). This reduction is however spurious: obviously, the estimation uncertainty seems to decrease if the data is extended. But since the uncertainty of the new data (obtained with bias correction) is not included in the analysis chain, this decrease of uncertainty has no direct value.

In conclusion, the answers of the authors to the reviewers' comments are not convincing and I thus reject this paper for publication in HESS.